# Computable universal online learning

**Dariusz Kalociński**
Institute of Computer Science
Polish Academy of Sciences
`d.kalocinski@ipipan.waw.pl`

**Tomasz Steifer**
Institute of Fundamental Technological Research
Polish Academy of Sciences
`tsteifer@ippt.pan.pl`

## Abstract

Understanding when learning is possible is a fundamental task in the theory of machine learning. However, many characterizations known from the literature deal with abstract learning as a mathematical object and ignore the crucial question: when can learning be implemented as a computer program? We address this question for universal online learning, a generalist theoretical model of online binary classification, recently characterized by Bousquet et al. (STOC´21). In this model, there is no hypothesis fixed in advance; instead, Adversary—playing the role of Nature—can change their mind as long as local consistency with the given class of hypotheses is maintained. We require Learner to achieve a finite number of mistakes while using a strategy that can be implemented as a computer program. We show that universal online learning does not imply computable universal online learning, even if the class of hypotheses is relatively easy from a computability-theoretic perspective. We then study the agnostic variant of computable universal online learning and provide an exact characterization of classes that are learnable in this sense. We also consider a variant of proper universal online learning and show exactly when it is possible. Together, our results give a more realistic perspective on the existing theory of online binary classification and the related problem of inductive inference.

## 1 Introduction

The quest to understand the fundamental limits of machine learning has led to the development of powerful theoretical frameworks, such as PAC learning or multi-armed bandits. These frameworks provide a high-level abstraction of machine learning, capable of capturing a broad class of learning problems within a single unified theoretical model. A good theoretical model should at least be able to say when learning is possible and when it is not. Take, for example, the online version of binary classification. The seminal papers by Littlestone (1988) and Angluin (1988) studied the question of when Learner has a strategy that guarantees a uniform bound on mistakes. Littlestone (1988) famously gave a combinatorial characterization—now known as the Littlestone dimension—of hypothesis classes that are learnable with a uniform mistake bound. Under such a restriction, one can equivalently assume that Adversary fixes one true function in advance. Recently, Bousquet et al. (2021) extended this work to a more general adversarial setting, where Learner makes finitely many mistakes (but possibly not uniformly bounded) and where Adversary can adapt dynamically. This setting is in line with real-world scenarios where uniform bounds are not always available. We call it *universal online learning.*[1]

The notion of the Littlestone dimension was generalized to ordinal numbers by Bousquet et al. (2021), who used it to give a characterization of universal online learnability: a class is universally online

---

[1] Please note that Bousquet et al. (2021) describe two related settings: one probabilistic (universal learning) and one adversarial (universal online learning). This paper is uniquely concerned with the latter (see Bousquet et al., 2020, Section 3).

39th Conference on Neural Information Processing Systems (NeurIPS 2025).

learnable if and only if it has an ordinal Littlestone dimension. More recently, the framework of universal online learning was extended to multiclass classification with bandit feedback (Hanneke et al., 2025).

Although the work of Bousquet et al. (2021) provides deep theoretical insights, it leaves open a critical question: when can such learning be implemented by an actual computer program? Having an appropriate Littlestone dimension is sufficient to guarantee the existence of a learner understood as an abstract object—a mathematical function. However, such a function may be computationally hard or even uncomputable, meaning that it cannot be implemented on a computer.

In this work, we bridge this gap by introducing computability constraints into the universal online learning framework, asking when a class of hypotheses admits a computable learning scheme that guarantees a finite number of mistakes.

Recently, different authors have begun to study combinatorial characterizations from the point of view of computability theory, introducing computable analogs of PAC learning (Agarwal et al., 2020; Sterkenburg, 2022; Delle Rose et al., 2023), robust PAC learning (Gourdeau et al., 2024), and uniform online learning (Hasrati and Ben-David, 2023). In all cases, special attention was given to the so-called recursively enumerably representable ($RER$) classes—classes specified by a program that enumerates all the functions in the class. In particular, $RER$ classes of finite Littlestone dimension are always learnable by a computable learner with a uniform mistake bound (Assos et al., 2023; Kozachinskiy and Steifer, 2024).

The standard notion of computability makes sense for countable domains such as the set of all natural numbers. For this reason, in this paper we limit our attention to universal online learning of hypotheses defined on $\mathbb{N}$. While this restriction may appear limiting, it is motivated by practical considerations. Indeed, all practical machine learning takes place on computers, which inherently operate over countable domains such as images or binary strings. From the point of view of computability, learnability on the natural numbers is sufficiently general to model any domain relevant to practical applications.[2]

## 2 Main results

We begin with a brief outline of the main results of the paper. This is followed by a detailed presentation of the framework, accompanied by examples and a discussion of the key concepts involved. The formal statements of all new theorems appear in Sections 2.1–2.3.

The first important observation (Section 2.1, Theorem 1) states that there exist $RER$ classes that are universally online learnable but cannot be learned by any computable learner. This stands in sharp contrast to the case of $RER$ classes with finite Littlestone dimension, as discussed in the introduction.

In Section 2.2, we investigate the agnostic variant of computable universal online learning. In agnostic learning, Adversary is no longer restricted to choose hypotheses that are consistent with the class. When the class is universally online learnable, one can guarantee a sublinear expected regret for agnostic learning (Littlestone and Warmuth, 1994; Cesa-Bianchi et al., 1997; Hutter et al., 2005; Blanchard et al., 2022; Hanneke et al., 2025). We show that this no longer holds when computability is required. In Theorem 3, we give an example of a class that is computably universally online learnable in the realizable case, but for which there cannot be a computable universal agnostic learner. On the positive side, Theorem 2 gives an exact characterization of classes that are agnostically online learnable by a computable learner. Furthermore, in Theorem 4, we show that the standard equivalence between realizable and agnostic learning holds if we restrict our attention to $RER$ classes.

Finally, in Section 2.3, we study a notion of *proper learning* adapted for universal online learning. Here, we say that a learner is proper if it outputs a realizable function while playing against realizable Adversary. Again, in Theorem 5 we give a precise characterization of proper learning in the computable regime. We also show that proper learning is harder than non-proper learning. In particular, Theorem 6 gives an example of a class for which computable universal online learners

---

[2]Despite our efforts to make the paper accessible to the machine learning community, the reader may occasionally wish to consult an introductory textbook on computability theory (Cutland, 1980), or a more advanced work (Odifreddi, 1989).

exist, but none of them can be proper in this sense. In fact, such examples can be found even among $RER$ classes.

## 2.1 Setting

Our model is given by the following game, played between Learner and Adversary. The game starts with a predefined hypothesis class $\mathcal{H}$, which is a set of (total) functions from some domain $X$ to $\{0, 1\}$. As announced in the introduction, $X = \mathbb{N}$. The game continues for infinitely many rounds. In each round, Adversary first gives Learner an element $x \in X$. Learner then makes a prediction $\hat{y} \in \{0, 1\}$ for the label of $x$. After the prediction, Adversary reveals the "true" label $y \in \{0, 1\}$ of $x$. If $y \neq \hat{y}$, we say that Learner has made a mistake.

Learner's goal is to make as few mistakes as possible. Otherwise, their moves are unrestricted in any other way. However, Adversary is expected to maintain consistency in their answers with $\mathcal{H}$. That means that at each step, the labels revealed so far must agree with some function $h \in \mathcal{H}$. Adversary is malicious in a very strong sense—they can change their mind; that is, they can change $h \in \mathcal{H}$ according to which their labels are given. An Adversary who maintains consistency with the class $\mathcal{H}$ is called *realizable*.

We use the term *learner* (without capitalization) to refer to a strategy according to which Learner could play. Formally, a *learner* is a partial function that maps elements of $(\mathbb{N} \times 0, 1)^{<\omega} \times \mathbb{N}$ to $\{0, 1\}$; namely, a function that, given a sample $S$ and a number $x$, predicts a binary label for $x$. A sequence $(x_t, y_t)_{t < N}$, where $N \in \mathbb{N} \cup \{\infty\}$, $x_t \in \mathbb{N}$, and $y_t \in \{0, 1\}$, is *realizable* by $\mathcal{H}$ (or $\mathcal{H}$-realizable) if, for every $T < N$, there exists $h \in \mathcal{H}$ such that $(x_t, y_t)_{t=0}^{T}$ is consistent with $h$, i.e., $h(x_t) = y_t$ for $t = 0, 1, \ldots, T$.

**Definition 1** (universal online learning). *We say that a class $\mathcal{H}$ is universally online learnable if there is a learner $L$ such that for every $\mathcal{H}$-realizable sequence $(x_t, y_t)_{t=0}^{\infty}$, there exists $m$ such that for every $n > m$ we have $L(S, x_{n+1}) = y_{n+1}$, where $S = (x_t, y_t)_{t=0}^{n}$. If this strategy can be implemented as a computer program, we say that $\mathcal{H}$ is computably universally online learnable.*

Equivalently, $\mathcal{H}$ is universally online learnable if there is a learner $L$ such that for every $\mathcal{H}$-realizable sequence $R$, there exists $M \in \mathbb{N}$ such that $L$ makes at most $M$ mistakes on $R$. In contrast, a class $\mathcal{H}$ is universally online learnable with a uniform mistake bound if there is a learner $L$ and a mistake bound $M \in \mathbb{N}$ such that for every $\mathcal{H}$-realizable sequence, $L$ makes at most $M$ mistakes on that sequence. Clearly, every class that is universally online learnable with a uniform mistake bound is also universally online learnable. The reverse does not hold (see Bousquet et al., 2020, Section 3.4).

In the uniform case, it does not matter whether Adversary is allowed to change the target function. If the Littlestone dimension is finite, a uniform mistake bound can be achieved even against adaptive Adversary. Conversely, if the Littlestone dimension is infinite, Adversary need not be adaptive to force mistakes on arbitrary sequences. For this reason, many authors prefer to assume that the target function is fixed in advance (see, e.g., Shalev-Shwartz and Ben-David, 2014). Indeed, this is also how the setting was originally defined by Littlestone (1988). The flexibility of Adversary becomes relevant only once the requirement of uniformity is dropped.

**Example 1.** Consider the class of all singletons, that is, the class of hypotheses on $\mathbb{N}$ that have exactly one nonzero label. Even though this class does not contain a function with all zeros, Adversary is allowed to play as if this were their "true" hypothesis. This is because every finite set of numbers all labeled with zeros is consistent with one of the functions in the class. This class is uniformly online learnable with a uniform mistake bound.

**Example 2.** Consider the class containing all the functions that have a finite number of nonzero labels. If Adversary cannot change their mind, this class is online learnable but with a non-uniform mistake bound. There are only countably many hypotheses, and we can check each one by one. However, in our setting, Adversary can effectively play whatever sequence of labels they want and still maintain consistency with the class. Therefore, this class is not universally online learnable.

**Example 3.** Imagine that the role of Adversary is played by a large language model (LLM). The generated sequence of tokens is not chosen in advance but produced online, and it is always consistent with the set of all possible outputs of this LLM. We can think about different random choices produced by the model as moves of Adversary. At a high level of abstraction, the theory of universal learning tells us when it is possible to predict (the binary representation of) this sequence of tokens with only a finite number of mistakes.

We will assume that our classes of hypotheses contain only total computable functions on the natural numbers, i.e., functions that can be implemented by a computer program which halts on every input. This is justified since computable universal online learning is only possible when all the hypotheses in the class are total computable (cf. Lemma 4). The classes of Examples 1–3 all contain only computable functions and all are $RER$, that is, there is a program that enumerates a list of programs computing all and only the functions from the class (see Definition 5). Some $RER$ classes are also decidably representable ($DR$), which means that they can be presented as a decidable set of programs. Not every $RER$ class is $DR$. The following classes are all $DR$ and computably universally online learnable (but not with a uniform mistake bound): all functions with finite support, thresholds on $\mathbb{N}$, and thresholds on $\mathbb{Z}$ (see Examples 3.8-3.10 in Bousquet et al., 2020).

The above examples motivate the notion of the closure $\overline{\mathcal{H}}$ of a class $\mathcal{H}$. The closure contains all functions that can be effectively used by Adversary who is realizable—that is, all the functions that are locally consistent with $\mathcal{H}$ after a finite number of steps.

**Definition 2.** *A function $g$ belongs to $\overline{\mathcal{H}}$ if there exists a sequence of functions $h_0, h_1, \ldots$ from $\mathcal{H}$ and a sequence of natural numbers $n_0 < n_1 < \ldots$ such that, for every $i \in \mathbb{N}$, $h_i(x) = g(x)$ for all $x < n_i$.*

Effectively, a mind-changing Adversary behaves as if they had selected a true hypothesis from $\overline{\mathcal{H}}$ in advance and then never changed their mind—this observation is straightforward to prove and is stated more precisely in Section 4.1.

It was recently discovered that any $RER$ class that is online learnable with a uniform mistake bound is also learnable by a computable learner (Assos et al., 2023; Kozachinskiy and Steifer, 2024). However, this implication fails when the requirement of a uniform mistake bound is dropped.

**Theorem 1.** *There exists a $RER$ class that is universally online learnable but not computably universally online learnable. In fact, such a class can even be $DR$.*

The proof of this result, given in Section 4.2, relies on the fact that even if $\mathcal{H}$ contains only computable functions, its closure may contain functions that are uncomputable. Similarly, natural examples of universally online learnable but not computably universally online learnable classes can be extracted from the existing literature (see, e.g., Cenzer and Remmel, 1998), where sets of solutions to a computably presented problem are represented as infinite paths in a computable tree. See Appendix D for details.

## 2.2 Agnostic learning

Our definition of computable universal online learning only asks that a computable learner behave in a certain way *on realizable samples*. In particular, this means that such a learner can be undefined on non-realizable samples. Think about a program that halts on realizable inputs but might go into an infinite loop if fed a different input. For instance, we can have a computable learner for the class of singletons from Example 1, with an additional instruction saying: once you see two numbers labeled with 1, go into a loop. This learner is still good in the realizable case, because Adversary will never show us a function with more than one nonzero label.

The situation changes once we try to move to the agnostic setting. We adopt a relatively weak requirement of sublinear regret, which is known to be equivalent to realizable universal online learning, even in the multiclass setting under bandit feedback (Hanneke et al., 2025). Such a property usually requires randomization. The notion of computable learning naturally generalizes to randomized computations. A randomized program is just a program that has access to an additional source of random bits sampled with probability $1/2$. We say that a randomized computable function $f$ is almost total if it is total with probability one with respect to its internal randomness. To be precise, with probability 1, once we fix the internal randomness, the computation halts on each input. We can now give the definition of universal agnostic online learnability. In the remainder of the paper, it is simply called *agnostic learnability*.

**Definition 3** (agnostic universal online learnability). *Let $L$ be a learner, and let $\mathcal{H}$ be a class of hypotheses. We say that $L$ agnostically universally online learns $\mathcal{H}$, or that $L$ is an agnostic universal*

*online learner for $\mathcal{H}$, if for every sequence $(x_t, y_t)_{t=0}^{\infty}$*

$$\mathbb{E}\left(\min_{h \in \mathcal{H}}\left(\sum_{0 < t \leq n} h(x_t) \neq y_t\right) - \sum_{0 < t \leq n}\left(L((x_i, y_i)_{i=0}^{t-1}, x_t) \neq y_t\right)\right)$$

*grows sublinearly in $n$, with the expectation taken relative to the internal randomness of the learner. Moreover, we say that $\mathcal{H}$ is computably agnostically universally online learnable if there exists a universal agnostic online learner for $\mathcal{H}$ which is realized as an almost total randomized computable function.*

**Example 4.** Consider a class of all decision trees of depth 3. In a realistic scenario, small decision trees cannot perfectly capture the complexity of the data. However, there are reasons for using small decision trees for classification, such as interpretability. Agnostic learning with small decision trees requires that, while we cannot perfectly classify the data, we can still do almost as well as the best decision tree.

In agnostic learning, Learner's strategy must be defined for all possible inputs, not only the realizable ones. In fact, the standard methods for transforming a realizable learner into an agnostic one require evaluating the (old) learner on non-realizable samples. Indeed, a standard approach is as follows: start with a method of prediction with expert advice (for a countable set of experts), like Weighted Majority Algorithm with nonuniform weights (Littlestone and Warmuth, 1994), Follow the Perturbed Leader (Hutter et al., 2005) or some version of Hedge algorithm (Blanchard et al., 2022) and apply it to the class of experts defined by simulating the realizable learner on different inputs.[3] In turn, having a computable agnostic learner we can transform it into a computable (non-randomized) realizable learner. The following theorem, proved in Section 4.3, formalizes this idea and gives two characterizations of agnostic learnability in the computable regime.

**Theorem 2.** *Let $\mathcal{H}$ be an arbitrary class of hypotheses. The following are equivalent:*

1. *$\mathcal{H}$ is computably universally online learnable (in the realizable setting) by a total learner.*

2. *There exists a $RER$ class $\mathcal{Z}$ such that $\overline{\mathcal{H}} \subseteq \mathcal{Z}$.*

3. *$\mathcal{H}$ is computably agnostically learnable.*

Without computability constraints, universal online learning implies agnostic learning (Hanneke et al., 2025). The following theorem shows that this implication does not hold in the computable regime. The proof is given in Section 4.4 and Appendix C.

**Theorem 3.** *There exists a class $\mathcal{H}$ that is computably universally online learnable but not computably agnostically learnable.*

The class given in the proof of Theorem 3 is not $RER$. It is not possible to computably enumerate all the hypotheses from this class. In Section 4.5 we prove that once we limit our attention to $RER$ classes, computable universal online learning is again equivalent to agnostic learning:

**Theorem 4.** *Let $\mathcal{H}$ be a $RER$ class. The following are equivalent:*

1. *$\mathcal{H}$ is computably universally online learnable.*

2. *There exists an $RER$ class $\mathcal{Z}$ such that $\overline{\mathcal{H}} \subseteq \mathcal{Z}$.*

3. *$\mathcal{H}$ is computably universally online learnable by a total learner.*

### 2.3 Proper versus improper learning

In computational learning theory, *proper learning* usually refers to the setting in which a learning algorithm is restricted to output hypotheses that belong to the class $\mathcal{H}$. In the uniform mistake-bound regime, this is equivalent to requiring that the learner output only realizable functions. In our setting, however, these two notions no longer coincide. The latter is too strong, since Adversary may effectively use a function that does not belong to the class but rather to its closure. Indeed, the standard definition can be satisfied only in the very restrictive case $\mathcal{H} = \overline{\mathcal{H}}$. Hence, we adopt the second interpretation, as it is the less trivial one.

---

[3]Ben-David et al. (2009) introduced this idea for the agnostic learning with respect to Littlestone classes.

**Definition 4** (proper learner). *We say that a learner $L$ is proper with respect to $\mathcal{H}$ if, for every $\mathcal{H}$-realizable sample $S$, $L(S, \cdot)$ is $\mathcal{H}$-realizable.*

In the above definition, $L(S, \cdot)$ is a total function $L' : \mathbb{N} \to \{0, 1\}$ of the single variable indicated by $\cdot$, defined as follows: $L'(x) = L(S, x)$, for all $x \in \mathbb{N}$.

When it is clear which class we are referring to, we often simply say that a learner is proper, instead of "proper with respect to ...". For example, in the formulation of the theorem below, "proper" means "proper with respect to $\mathcal{H}$". The following characterization is proved is Section 4.6:

**Theorem 5.** *For every $RER$ class $\mathcal{H}$, $\mathcal{H}$ is computably universally online learnable by a proper learner if and only if $\overline{\mathcal{H}}$ is a $RER$ class.*

Trivially, if a class is computably universally online learnable by a proper learner, then it is computably universally online learnable. In Section 4.7, we show that the reverse implication does not hold, thereby establishing a separation between proper computable universal online learning and computable universal online learning.

**Theorem 6.** *There exists a $RER$ class that is computably universally online learnable but has no computable proper learner.*

## 3 Discussion

Theorems 2 and 5 provide an exact characterization of agnostic and proper learning in the framework of universal online learning. Together with Theorem 4, this yields an exact characterization of realizable universal online learning for the family of $RER$ classes, which arguably includes all hypothesis classes of practical relevance. It remains an interesting open problem to provide an exact characterization of realizable computable universal online learning for arbitrary classes. A related open question is how to characterize computable online learning with a uniform mistake bound. A recent attempt by Delle Rose et al. (2025) suggests that this question may be hard.

Theorem 1, together with the accompanying discussion, indicates that universally online learnable $RER$ classes without computable learners do exist and can arise in naturally occurring problems. How complex can the learners for such classes be? This question admits a precise computability-theoretic formulation, and its resolution will be presented in a separate manuscript (in preparation).

## 4 Proofs

For the sake of mathematical rigor, we state certain notions in a more precise way. Programs in a fixed programming language are finite objects and can be indexed by natural numbers. We use the notation $\varphi_n$ to denote the function computed by the program with index $n$ ($n$-th program, or program $n$). Such a function can be partial, that is, undefined on some inputs. We say that a set $Z \subseteq \mathbb{N}$ is computably enumerable (c.e.) if there exists a program that enumerates all (and only) the elements of $Z$. If $z_0, z_1, \ldots$ is such an enumeration of $Z$ generated by a program, we call it a computable enumeration of $Z$. In particular, $Z$ can be a set of (indices of) programs.

**Definition 5** ($RER$ class). *$\mathcal{H}$ is $RER$ if there exists a c.e. set $H$ (of the indices of programs) such that, for each total function $f$ from $\mathbb{N}$ to $\{0, 1\}$, $f \in \mathcal{H}$ iff there exists $e \in H$ such that $\varphi_e = f$.*

Equivalently, $\mathcal{H}$ is $RER$ if there exists a c.e. set $H$ such that $\mathcal{H} = \{\varphi_e : e \in H\}$.

### 4.1 Helpful lemmas

Before we dive into the proof of the main results, it will be helpful to state some technical lemmas. The proofs of these observations use standard techniques, and we relegate them all to Appendix A.

In our setting, Adversary can ask Learner to label numbers in an arbitrary order, perhaps even omitting some of the numbers completely. Adversary has to maintain consistency of each sample with some hypothesis in the class but we imagine that at each round there might be a different target hypothesis. Lemmas 1 and 2 tell us that equivalently we can assume that Adversary just chooses in advance a single target function from the closure of the class.

**Lemma 1.** *Let $h \in \overline{\mathcal{H}}$ and let $(x_t)_{t=0}^{\infty}$ be any sequence of non-negative integers. Then the sequence $(x_t, h(x_t))_{t=0}^{\infty}$ is $\mathcal{H}$-realizable.*

**Lemma 2** (Closure Lemma)**.** *Every $\mathcal{H}$-realizable sequence $(x_i, y_i)_{i=0}^{\infty}$ is consistent with some element of $\overline{\mathcal{H}}$. In particular, if an $\mathcal{H}$-realizable sequence $(x_i, y_i)_{i=0}^{\infty}$ is such that the enumeration $x_0, x_1, \ldots$ mentions all natural numbers, then the function $x_i \mapsto y_i$ is an element of $\overline{\mathcal{H}}$.*

The next lemma states that if $\mathcal{H}$ is learnable even in a very weak sense, then there exists a sample on which the learner "outputs" a given hypothesis from $\overline{\mathcal{H}}$. This lemma describes a phenomenon similar to *locking sequences* from the inductive inference theory, first considered by Blum and Blum (1975).

**Lemma 3** (Projection Lemma)**.** *Fix a class of hypotheses $\mathcal{H}$ and a learner $L$. Suppose that for each $\mathcal{H}$-realizable play of Adversary, $L$ predicts infinitely many labels correctly, i.e., for each $h \in \overline{\mathcal{H}}$ and $X = ((x_1, h(x_1)), (x_2, h(x_2)), \ldots)$ there are infinitely many $k$ such that $L(S, x_{k+1}) = h(x_{k+1})$, where $S$ is the prefix of $X$ of length $k$. Then there exists a sample $S'$ such that $S'$ is consistent with $h$ and $L(S', \cdot) = h$; moreover, such a sample $S'$ is always $\mathcal{H}$-realizable.*

The above lemma implies the following:

**Lemma 4.** *Let $\mathcal{H}$ be computably universally online learnable. Then each member of $\overline{\mathcal{H}}$ is computable.*

## 4.2 Proof of Theorem 1

**Theorem 1.** *There exists a $RER$ class that is universally online learnable but not computably universally online learnable. In fact, such a class can even be $DR$.*

*Proof.* A tree is a set of strings $T$ such that, if $\sigma \in T$, then each prefix of $\sigma$ is also a member of $T$. A tree is computable if there is a program that, for a given $\sigma$, decides whether $\sigma$ belongs to the tree. An infinite path in $T$ is an infinite string such that each its prefixes is a member of $T$. We denote the set of infinite paths in $T$ by $[T]$. It is a folklore result that there exist computable binary trees with at most countably many infinite paths, at least one of which is uncomputable (see, e.g., Cenzer et al., 1986). Fix such a tree $T$. Let $\mathcal{H} = \{\sigma 0^{\infty} : \sigma \in T\}$. It is easy to see that $\mathcal{H}$ is a $RER$ class, $[T] \subseteq \overline{\mathcal{H}}$ and that $\overline{\mathcal{H}}$ is countable. Given an enumeration $h_1, h_2, \ldots$ of $\overline{\mathcal{H}}$, an uncomputable learner for $\mathcal{H}$ can be defined as follows: given a sample $S$ and $x$, we choose the least $i$ such that $h_i$ is consistent with $S$ and we output $h_i(x)$. However, there is no computable learner for $\mathcal{H}$. Otherwise, by Lemma 4, every member of $\overline{\mathcal{H}}$ would be computable, and thus all paths of $T$ would be computable. $\qquad\square$

## 4.3 Proof of Theorem 2

**Theorem 2.** *Let $\mathcal{H}$ be an arbitrary class of hypotheses. The following are equivalent:*

1. *$\mathcal{H}$ is computably universally online learnable (in the realizable setting) by a total learner.*

2. *There exists a $RER$ class $\mathcal{Z}$ such that $\overline{\mathcal{H}} \subseteq \mathcal{Z}$.*

3. *$\mathcal{H}$ is computably agnostically learnable.*

*Proof.* The implication (1) $\Rightarrow$ (2) follows from Lemma 3. Assume that $\mathcal{H}$ is computably universally online learnable via a total learner $L$. Let $S_0, S_1, \ldots$ be a computable enumeration of all possible samples (not necessarily realizable ones). For each $S_n$, the function $L(S_n, \cdot)$ is total computable with range $\{0, 1\}$. This gives us a computable enumeration of the class $\mathcal{Z} = \{L(S_n, \cdot) : n \in \mathbb{N}\}$, which shows that $\mathcal{Z}$ is $RER$. Lemma 3 then allows us to conclude that $\mathcal{Z}$ is a superset of $\overline{\mathcal{H}}$.

The implication (2) $\Rightarrow$ (3) follows from the standard results about prediction with expert advice using countable experts. To be precise, we use the following lemma:

**Lemma 5.** *Given a computably enumerable set of total computable functions $A$, there exists a computable randomized algorithm with sublinear regret relative to the best function in $A$.*

Several techniques are known that can be used to obtain the above lemma (see, e.g., Theorem 5 in Hutter et al., 2005). Computability of these techniques is usually only briefly discussed or even not mentioned at all. We give a detailed proof of computability for one such algorithm in the Appendix B.

Finally, we argue how to transform a computable agnostic learner into a total computable universal for the realizable case. The argument proceeds in two steps. First, we show that $(3) \Rightarrow (2)$. Then, the implication $(2) \Rightarrow (1)$ is straightforward: the strategy of the learner is to enumerate the elements of $Z$ and try each hypothesis until we stop making mistakes.

We now argue that $(3) \Rightarrow (2)$. Again, we want to apply Lemma 3 to show that we can obtain an enumeration of a superset of $\overline{\mathcal{H}}$ by enumerating all functions of the form $L'(S, \cdot)$, where $L'$ satisfies the condition from Lemma 3 (i.e., making infinitely many correct predictions against any Adversary). Therefore, we need to show how to transform the randomized agnostic learner $L$ into a deterministic $L'$ that makes infinitely many correct predictions on any $\mathcal{H}$-realizable sequence.

In principle, it suffices to consider the majority vote taken over the internal randomness of $L$, that is, to always predict the more probable outcome (breaking ties arbitrarily). However, the majority vote for an almost total randomized function is not always computable. To guarantee computability, we need one additional simple trick. Imagine that we run an almost total randomized algorithm $f$ on some input $x$, and the computation stops after a finite number of steps. In particular, this means that $f$ has seen only a finite number of bits of the random advice—$f$ can read at most one bit at each computation step. Now, suppose that we simulate $f(x)$ for all possible random advices until the computations stop for at least $1 - \epsilon$ probability mass of the random advices, for some small $\epsilon > 0$. This happens after a finite number of steps since $f$ is defined on random advices of probability one.

With that in mind, we define the learner $L'$. Given the $k$-th number to label, $L'$ first simulates the behavior of $L$ on all random inputs until these computations stop for at least $1 - (k+1)^{-2}$ probability measure of random advices. For each round, this also induces a new probability measure on random advices, where the whole probability mass is restricted to the set of the uniform measure $1 - (k+1)^{-2}$. Let us denote the product of these probability measures by $P'$.

Now, suppose that we are playing against Adversary on some $\mathcal{H}$-realizable sequence $X$. Let us define a new random variable $\eta(S)$ as the number of mistakes made by $L$ on a sample $S$ divided by $|S|$. Also, let $X_n$ be the prefix of $X$ of length $n$. Since $L$ is an agnostic learner for $\mathcal{H}$, we have $\limsup_{n \to \infty} \mathbb{E}(\eta(X_n) = 0$. By the dominated convergence theorem, we obtain $\mathbb{E}(\limsup_{n \to \infty} \eta(X_n) = 0)$. At the same time, we can observe that the same property holds if the expectation is taken with respect to the $P'$ measure. This is because, at each round, the probability of making an error cannot increase more than $(k+1)^{-2}$ (recall that for binary variables expectation is equal to the probability of sampling 1). Since $\sum_{k>0}(k+1)^{-2}$ converges, this additional error can be neglected in the limit.

This allows us to apply Markov's inequality to conclude that $\limsup_{n \to \infty} \eta(X_n) > 0$ occurs with $P'$-probability at least $1/2$. As a consequence, $L'$ makes infinitely many correct predictions on $X$. Since $X$ was chosen to be an arbitrary realizable sequence, we have shown that $L'$ satisfies the condition from Lemma 2. $\qquad \square$

## 4.4 Proof of Theorem 3

**Theorem 3.** *There exists a class $\mathcal{H}$ that is computably universally online learnable but not computably agnostically learnable.*

*Proof.* We define a hypothesis class $\mathcal{H}$ that is computably universally online learnable yet has no total computable learner. Once this is established, the result follows by an application of Theorem 2.

Recall that $\varphi_n$ denotes the function computed by the $n$-th program. We say that an infinite binary string $h$ is an *evil sequence* for a learner $\varphi_n$ if it satisfies the following conditions:

1. The initial segment of $h$ of length $n + 1$ is $0^n 1$; i.e., $h {\restriction} (n + 1) = 0^n 1$;

2. For every $k > n$, when the learner $\varphi_n$ is given the first $k$ bits of $h$—that is, the sample $(i, h(i))_{i=0}^{k-1}$—and is asked to predict the $k$-th bit, it outputs $1 - h(k)$.

If $\varphi_n$ is a total learner, then there is a unique evil sequence for $\varphi_n$. We define $\mathcal{H}$ as the least class containing the evil sequences of all total learners $\varphi_n$.

Clearly, no total computable $\varphi_n$ learns $\mathcal{H}$, because it makes infinitely many mistakes on the realizable sequence $(i, h(i))_{i=0}^{\infty}$, where $h$ is the evil sequence for $\varphi_n$.

Before presenting a computable learner for $\mathcal{H}$, we need to establish certain properties of the class. Countability follows from the fact that there are countably many total learners $\varphi_n$, and each of them uniquely determines an evil sequence. Below, we show that $\overline{\mathcal{H}} = \mathcal{H} \cup \{0^\infty\}$.

It is clear that $0^\infty \in \overline{\mathcal{H}}$ because there are hypotheses in $\mathcal{H}$ that start with $0^n$, for infinitely many $n$. Towards a contradiction, suppose that $f \in \overline{\mathcal{H}} \setminus \mathcal{H}$ and $f \neq 0^\infty$. Then there exists a least $n_0$ with $f(n_0) = 1$. Hence, some hypothesis $h \in \mathcal{H}$ starts with $0^{n_0}1$. By the uniqueness of evil sequences, it follows that $\mathcal{H}$ contains only one hypothesis extending $0^{n_0}1$. However, the assumption $f \in \overline{\mathcal{H}} \setminus \mathcal{H}$ implies that there are natural numbers $n_1 < n_2 < \ldots$ and hypotheses $h_1, h_2, \ldots$ from $\mathcal{H}$ such that each $h_i$ is distinct from $f$ but $h_i$ and $f$ agree on the first $n_i$ bits. It follows that $\mathcal{H}$ contains infinitely many hypotheses extending $0^{n_0}1$, a contradiction.

A partial function $e(n, x)$ such that, for every $n$, if $\varphi_n$ is a total learner, then $e(n, \cdot)$ is the evil sequence for $\varphi_n$, can be computed by the following well-defined recursive procedure: on input $x \leq n$, output the $x$-th bit of $0^n1$; on input $x > n$, output $\hat{y} \in \{0, 1\}$ such that $\hat{y} \neq \varphi_n((i, e(n, i))_{i=0}^{x-1}, x)$.

**Computable learner for $\mathcal{H}$.** We start by guessing according to the hypothesis $0^\infty$ until Adversary reveals a one. Suppose the latter happens for the first time at position $m$, i.e., $y(m) = 1$. At this point we know that Adversary is restricted to finitely many hypotheses from $\mathcal{H}_m := \{h \in \mathcal{H} : h$ is the evil sequence for a total learner $\varphi_n, n \leq m\}$. At subsequent stages, given a sample $S$ and an element $x$, we search for $n \leq m$ such that $e(n, 0), \ldots, e(n, M)$ are defined and consistent with $S$, where $M$ is the maximum of $x$ and all numbers appearing in $S$. Such $n$ exists, because Adversary is consistent with an element of $\mathcal{H}_m$ and $e$ is able to compute it, as we observed above. Once such $n$ is found, we predict $e(n, x)$. Notice that if we make a mistake by predicting $e(n, x)$, we will not predict according to $e(n, \cdot)$ again. If we were to make infinitely many mistakes, it would mean that $\mathcal{H}_m$ is infinite, which is not the case. $\qquad\square$

## 4.5 Proof of Theorem 4

**Theorem 4.** *Let $\mathcal{H}$ be a RER class. The following are equivalent:*

1. *$\mathcal{H}$ is computably universally online learnable.*

2. *There exists an RER class $\mathcal{Z}$ such that $\overline{\mathcal{H}} \subseteq \mathcal{Z}$.*

3. *$\mathcal{H}$ is computably universally online learnable by a total learner.*

*Proof.* Let $\mathcal{H}$ be a $RER$ class. The implication $(3) \Rightarrow (1)$ is obvious. We first prove $(1) \Rightarrow (2)$. Assume that $\mathcal{H}$ is computably universally online learnable via $L$. Let $S_0, S_1, \ldots$ be a computable enumeration of all finite $\mathcal{H}$-realizable samples. Such an enumeration exists because $\mathcal{H}$ is a $RER$ class. For each $S_n$, the function $L(S_n, \cdot)$ is total computable with range $\{0, 1\}$. This follows from the fact that $S_n$ is $\mathcal{H}$-realizable and that $L$ is a learner for $\mathcal{H}$. We observe that the class $\mathcal{Z} := \{L(S_n, \cdot) : n \in \mathbb{N}\}$ is $RER$. The reason for this is as follows: each $L(S_n, \cdot)$ is a function computable by a program that can be effectively obtained from the program computing $L$ and $n$, uniformly in $n$. (We use this kind of reasoning frequently, but state it explicitly only here.)

It remains to prove that $\overline{\mathcal{H}} \subseteq Z$. Let $h \in \overline{\mathcal{H}}$. By Lemma 3, we can choose an $\mathcal{H}$-realizable sample $S_e$ such that $S_e$ is consistent with $h$ and $L(S_e, \cdot) = h$. Therefore, $h \in \mathcal{Z}$.

Now, we prove $(2) \Rightarrow (3)$. Suppose $\overline{\mathcal{H}} \subseteq \mathcal{Z}$, for some $RER$ class $\mathcal{Z}$. Let $Z$ be a c.e. set of programs such that $\mathcal{Z} = \{\varphi_e : e \in Z\}$, and let $z_0, z_1, \ldots$ be a computable enumeration of $Z$. We define a learner $L$ for $\mathcal{H}$ as follows. On input $(S, x)$, where $S$ is any (finite) sample and $x \in \mathbb{N}$, we compute $i_0 :=$ the least $i < |S|$ such that $S$ is consistent with $\varphi_{z_i}$. If no such $i < |S|$ exists, we set $i_0 = |S|$. The output of the learner on $(S, x)$ is $\varphi_{z_{i_0}}(x)$. Since each $\varphi_{z_i}$ is total, $i_0$ and $\varphi_{z_{i_0}}(x)$ are always defined. Therefore, $L$ is total computable.

To show that $L$ learns $\mathcal{H}$, let $R = (x_t, y_t)_{t=0}^\infty$ be $\mathcal{H}$-realizable. By Lemma 2, and the fact that $\overline{\mathcal{H}} \subseteq \mathcal{Z}$, there exists $j_0 :=$ the least $j$ such that for all $n$, $R_{<n}$ is consistent with $\varphi_{z_j}$. By the definition of $L$, for almost all $n$, $L(R_{<n}, x_n) = \varphi_{z_{j_0}}(x_n)$, and thus $L$ makes only finitely many mistakes on $R$, hence $\mathcal{H}$ is computably universally online learnable by a total learner. $\qquad\square$

### 4.6 Proof of Theorem 5

**Theorem 5.** *For every $RER$ class $\mathcal{H}$, $\mathcal{H}$ is computably universally online learnable by a proper learner if and only if $\overline{\mathcal{H}}$ is a $RER$ class.*

*Proof.* Let $\mathcal{H}$ be $RER$. To prove ($\Rightarrow$), assume that $\overline{\mathcal{H}}$ is $RER$, take $\mathcal{Z} := \overline{\mathcal{H}}$, and proceed as in the proof of the implication (2) $\Rightarrow$ (3) in Theorem 4. This way, we obtain a computable proper learner for $\mathcal{H}$.

To prove ($\Leftarrow$), assume that $\mathcal{H}$ is computably universally online learnable by a proper learner $L$. Let $S_0, S_1, \ldots$ be a computable enumeration of all $\mathcal{H}$-realizable samples. Let $\mathcal{F} = \{L(S_n, \cdot) : n \in \mathbb{N}\}$. Clearly, $\mathcal{F}$ is $RER$. Below, we show that $\mathcal{F} = \overline{\mathcal{H}}$.

To show $\mathcal{F} \subseteq \overline{\mathcal{H}}$, let $g \in \mathcal{F}$. We have $g = L(S_n, \cdot)$, for some $n$. By the definition of proper learner (Definition 4), $g$ is $\mathcal{H}$-realizable. By Lemma 2, it follows that $g \in \overline{\mathcal{H}}$.

To prove that $\overline{\mathcal{H}} \subseteq \mathcal{F}$, let $g \in \overline{\mathcal{H}}$. By Lemma 3, there exists $n$ such that $S_n$ is consistent with $g$ and $L(S_n, \cdot) = g$. Hence, by the definition of $\mathcal{F}$, we have $g \in \mathcal{F}$. $\qquad\square$

### 4.7 Proof sketch of Theorem 6

**Theorem 6.** *There exists a $RER$ class that is computably universally online learnable but has no computable proper learner.*

The full proof can be found in Appendix C. We construct a class $\mathcal{A}$ as stated in the theorem. The construction of $\mathcal{A}$ is a program that enumerates its elements. To ensure that no computable learner for $\mathcal{A}$ is proper, we satisfy the following requirement $R_e$ for each program $e$:

$$\text{If a learner } \varphi_e \text{ is proper with respect to } \mathcal{A}, \text{ then } \varphi_e \text{ does not learn } \mathcal{A}. \qquad (R_e)$$

Below, we briefly explain the approach to satisfying the requirements. Each $R_e$ is assigned its unique starting string $0^e 1$. The strategy for the requirement $R_e$ operates at timesteps $s \geq e$ and is allowed to put into $\mathcal{A}$ only certain elements of $[0^e 1]$. By definition, $[0^e 1]$ consists of all infinite binary sequences that extend $0^e 1$.

**Strategy for $R_e$.** At timestep $s = e$, there is only one point in $[0^e 1] \cap \mathcal{A}$, namely $0^e 1 0^\infty$. We let $\gamma_e := 0^e 1$. At subsequent timesteps, we monitor which bit $\varphi_e$ predicts to come right after $\gamma_e$, knowing that currently in $\mathcal{A}$ there is only one element extending $\gamma_e$, namely $\gamma_e 0^\infty$. More precisely, we think of $\gamma_e$ as the sample $((0, \gamma_e(0)), (1, \gamma_e(1)), \ldots, (l-1, \gamma_e(l-1)))$, where $l$ is the length of $\gamma_e$, and we monitor the value of $\varphi_e(\gamma_e, l)$. Obviously, we cannot know this in advance—we can only check, at subsequent timesteps of the construction, whether the program $e$ outputs the prediction before executing at most $s$ instructions.

If $e$ never (that is, at no timestep $s$) outputs a prediction, then $e$ is not a learner for $\mathcal{A}$, because it is undefined on the $\mathcal{A}$-realizable sample $\gamma_e$. In this case, $R_e$ is vacuously satisfied, since the antecedent of $R_e$ is false. Now suppose that $e$ eventually outputs a prediction. Then we consider two cases.

In the first case, we see that $\varphi_e$ is not proper for the current $\mathcal{A}$—that is, $\varphi_e(\gamma_e, l) = 1$ (recall that, currently, the only element of $\mathcal{A}$ extending $\gamma_e$ is $\gamma_e 0^\infty$). In that case, we stop enumerating any new points from $[0^e 1]$ into $\mathcal{A}$. Consequently, $\varphi_e$ will remain non-proper for $\mathcal{A}$ and $R_e$ will be satisfied forever.

In the second case, we see that $\varphi_e$ may be proper for the current $\mathcal{A}$—that is, $\varphi_e(\gamma_e, l) = 0$. In that case, we enumerate into $\mathcal{A}$ a new hypothesis $\gamma_e 1 0^\infty$. This way, we force $\varphi_e(\gamma_e, l)$ to make a mistake, because now Adversary could disagree with the prediction $\varphi_e(\gamma_e, l) = 0$ and reply with 1. We set $\gamma_e := \gamma_e 1$ and $l := |\gamma_e|$, and the strategy continues monitoring the prediction of $\varphi_e(\gamma_e, l)$, with these new values of $\gamma_e, l$. This ends the description of the strategy for $R_e$.

If, at subsequent stages, we keep getting the result that $\varphi_e(\gamma_e, l)$ is proper for the current $\mathcal{A}$ (the second case above), we will put into $[0^e 1] \cap \mathcal{A}$ infinitely many hypotheses (in the limit), namely $0^e 1^n 0^\infty$, for every $n > 0$. By the construction, $\varphi_e$ will make infinitely many mistakes on the $\mathcal{A}$-realizable sequence $0^e 1^\infty$. Consequently, $\varphi_e$ will not be a computable learner for $\mathcal{A}$ and $R_e$ will be satisfied.

## Acknowledgments and Disclosure of Funding

Dariusz Kalociński acknowledges support of National Science Centre Poland grant under the agreement no. 2023/49/B/HS1/03930. Tomasz Steifer was supported by the Agencia Nacional de Investigación y Desarrollo grant no. 3230203.

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

# A   Proofs of helpful lemmas

To prove the required lemmas from Section 4.1, we invoke basic concepts and results from topology.

The Cantor space is the topological space on $2^\omega$ (the set of all infinite binary sequences) with basic open sets of the form $[\sigma] := \{\sigma X : X \in 2^\omega\}$ for all $\sigma \in 2^{<\omega}$ (finite binary strings). For brevity, we denote this space by $2^\omega$. It is clear that members of $2^\omega$ correspond to hypotheses, and that sets of points in the Cantor space correspond to hypothesis classes. The notion of closure introduced in the main text coincides with the standard topological closure: the closure of $\mathcal{A}$, denoted by $\overline{\mathcal{A}}$, is defined as the least closed set containing $\mathcal{A}$.

We say that $p$ is an accumulation point of $\mathcal{A}$ if $p \in \overline{\mathcal{A} \setminus \{p\}}$. In other words, every neighborhood of $p$ contains points of $\mathcal{A}$ different from $p$. Intuitively, $p$ can be "approximated" by points in $\mathcal{A}$. The set of accumulation points of $\mathcal{A}$ is denoted by $D(\mathcal{A})$ and is called the *derived set* of $\mathcal{A}$. The closure of a class can then be expressed in terms of the derived set as $\overline{\mathcal{A}} = \mathcal{A} \cup D(\mathcal{A})$.

*Proof of Lemma 1.* The lemma is obvious if $h \in \mathcal{H}$. Now suppose $h \in \overline{\mathcal{H}} \setminus \mathcal{H}$. Since $\overline{\mathcal{H}} = \mathcal{H} \cup D(\mathcal{H})$, it follows that $h \in D(\mathcal{H})$. Therefore, every open set containing $h$ also contains points of $\mathcal{H}$ distinct from $h$ (see, e.g., Kuratowski, 1972, Theorem 1, p. 141). In other words, for every $n \in \mathbb{N}$, there exists $h_n \in \mathcal{H}$ such that $h \neq h_n$ and $h_n \in [h \upharpoonright n]$. It follows that for each $n$, $(x_t, h(x_t))_{t=0}^{n-1}$ is consistent with $h_n \in \mathcal{H}$. Hence, $h$ is $\mathcal{H}$-realizable. $\qquad\square$

Lemma 2 follows from some basic properties the Cantor space. Recall that the Cantor space is induced by the following metric on $2^\omega$, known as the Baire metric:

$$\delta(f, g) = \begin{cases} 0 & \text{if } f = g, \\ \frac{1}{\mu x[f(x) \neq g(x)]} & \text{otherwise.} \end{cases} \tag{1}$$

The Cantor space is complete with respect to $\delta$.

*Proof of Lemma 2.* Let $s = (x_i, y_i)_{i=0}^\infty$ be an $\mathcal{H}$-realizable sequence. Hence, there exists a sequence $(h_i)_{i=0}^\infty$ of elements of $\mathcal{H}$ such that for every $k \in \mathbb{N}$, the sample $(x_i, y_i)_{i=0}^k$ is consistent with $h_k$. We call such a sequence a witness.

Suppose there exists a witness $(h_i)_{i=0}^\infty$ such that $\{h_0, h_1, \dots\}$ is finite. Then there is $\zeta \in \mathcal{H}$ that occurs in $(h_i)_{i=0}^\infty$ infinitely often, and therefore $\zeta$ is consistent with $(x_i, y_i)_{i=0}^\infty$. Moreover, $\zeta \in \mathcal{H} \subseteq \overline{\mathcal{H}}$, so $\zeta$ is as desired.

Now suppose that for every witness $(h_i)_{i=0}^\infty$, the set $\{h_0, h_1, \dots\}$ is infinite. Fix a witness $(h_i)_{i=0}^\infty$ and let $W = \{h_0, h_1, \dots\}$.

Let $D = \{x_0, x_1, \dots\}$. We define a sequence of subsets of $2^\omega$. We set $C_{-1} = 2^\omega$. For $n \geq 0$, we define:

$$C_n = \begin{cases} C_{n-1} \cap [(n, s(n))] & \text{if } n \in D, \text{ and} \\ C_{n-1} \cap [(n, b)] & \text{if } n \notin D, b \in \{0, 1\} \text{ and } C_{n-1} \cap [(n, b)] \cap W \text{ is infinite.} \end{cases} \tag{2}$$

Above, $[(n, s(n))] = \{X \in 2^\omega : X(n) = s(n)\}$.

It is clear that $C_{-1} \supseteq C_0 \supseteq C_1 \supseteq \dots$. We now show that $C_n \cap W$ is infinite, for $n = -1, 0, 1, 2, \dots$. This is obvious for $n = -1$. Fix $n \geq 0$ and suppose that $C_{n-1} \cap W$ is infinite.

First, assume $n \in D$. By the definition of a witness, we see that $[n, s(n)] \cap W \supseteq \{h_j \in \mathcal{H} : j \geq n\}$. Since $C_{n-1} \cap W$ is infinite, $C_n \cap W = C_{n-1} \cap [n, s(n)] \cap W$ is also infinite.

Second, assume that $n \notin D$. Now, since $C_{n-1} \cap W$ is infinite, it must be the case that either $C_{n-1} \cap W \cap [(n, 0)]$ or $C_{n-1} \cap W \cap [(n, 1)]$ is infinite. So we can choose $b$ that satisfies (2).

Observe that $C_{-1}, C_0, C_1, \dots$ are closed. Obviously, $C_{-1}$ is closed, and if $C_{n-1}$ is closed then $C_n$, being the intersection of $C_{n-1}$ and a clopen set, is also closed.

Pick a sequence of points $(\zeta_n)_{n=-1}^\infty$ such that $\zeta_n \in C_n \cap W$. The sequence $(\zeta_n)_{n=-1}^\infty$ is Cauchy under the Baire metric. Since the Cantor space is complete with respect to the Baire metric, there exists $\lim_{n \to \infty} \zeta_n = \zeta \in 2^\omega$. All elements of $(\zeta_n)_{n=-1}^\infty$ belong to $\mathcal{H}$. Since $\overline{\mathcal{H}}$ is closed, $\zeta \in \overline{\mathcal{H}}$.

It remains to show that $\zeta$ is consistent with $s$. Suppose not. Hence, for some $x \in D$, $\zeta(x) \neq s(x)$. Hence $\zeta \in [(x, 1 - s(x))]$, so $\zeta \notin C_x$. On the other hand, for each $n = -1, 0, 1, \ldots$, almost all elements of $(\zeta_n)_{n=-1}^{\infty}$ are in $C_n$, and since $C_n$ is closed, $\zeta \in C_n$. So $\zeta \in C_x$ and $\zeta \notin C_x$, a contradiction. So $\zeta$ is consistent with $s$. $\qquad\square$

*Proof of Lemma 3.* Let $\mathcal{H}$ be a class of hypotheses and $L$ a learner. Assume that for each $\mathcal{H}$-realizable play of the Adversary, $L$ predicts infinitely many labels correctly. Towards a contradiction, suppose that there exists $h \in \overline{\mathcal{H}}$ such that for every sample $S$, if $S$ is consistent with $h$ then $L(S, \cdot) \neq h$. Choose such an $h$.

We define a sequence $(U_n)_{n=0}^{\infty}$ by induction. Let $U_0 = \emptyset$. Given $U_n$, we define $U_{n+1}$ as $U_n$ prolonged by $(x_n, h(x_n))$, where $x_n$ is the least $x$ such that $L(U_n, x) \neq h(x)$. This definition is correct which can be shown by a simple inductive argument.

Each $U_n = (x_t, h(x_t))_{t=1}^{n}$ is an initial segment of the sequence $(x_t, h(x_t))_{t=1}^{\infty}$. By Lemma 1, $(x_t, h(x_t))_{t=1}^{\infty}$ is $\mathcal{H}$-realizable so by assumption, $L$ predicts infinitely many labels correctly when playing against $(x_t, h(x_t))_{t=1}^{\infty}$. However, $L(U_n, x_{n+1}) \neq h(x_{n+1})$, for all $n$. A contradiction. This proves the first part of the lemma.

The second part follows from the simple observation that if $S$ is consistent with $h \in \overline{\mathcal{H}}$ then $S$ is $\mathcal{H}$-realizable. $\qquad\square$

*Proof of Lemma 4.* Let $L$ be a computable learner for $\mathcal{H}$ and let $h \in \overline{\mathcal{H}}$. By Lemma 3, there exists a sample $S$ such that $L(S, \cdot) = h$. Hence, $h$ is computable. $\qquad\square$

# B   Proof of Lemma 5

Suppose that we have a countable set of experts $N = \{e_1, e_2, \ldots\}$. Our goal is to prove that there exists an almost total randomized computable algorithm that achieves sublinear expected regret with respect to $N$ (where the expectation is taken with respect to the random advice sampled from the uniform measure). We will employ a variant of Weighted Majority Algorithm ($WMA$), also known as Exponentially Weighted Average (Littlestone and Warmuth, 1994; Cesa-Bianchi et al., 1997), combined with the so-called *doubling trick* and increasing the number of experts over time.

First, we briefly recall the randomized version of the algorithm for a fixed time horizon $T$ and a finite number of experts $n$. At each round $t \leq T$, the algorithm computes the weight $w_i(t)$ for each expert $e_i$. The prediction is obtained by sampling from a probability distribution supported on the set of experts, with the probability of each expert $e_i$ being $w_i(t)/\sum_{j \leq n} w_j(t)$, and returning the prediction of the chosen expert. The weights are initialized as uniform ($w_i(1) = 1$) and then, at each round, we multiply the weight of the expert by $e^\eta$ if and only if the expert made an incorrect prediction in the last round (otherwise, the weight remains the same). More precisely, $w_i(t) = e^{\eta L_i(t)}$, where $L_i(t)$ is the number of incorrect predictions of expert $e_i$ before round $t$. It is known that for the appropriate value of $\eta$, depending in a computable way on $T$ and $n$, the expected regret of this algorithm with respect to the finite set of experts, at each round $t \leq T$, is bounded by $\sqrt{\frac{n}{2} \ln T}$ (see Theorem 2.2. in Cesa-Bianchi and Lugosi (2006) for a modern exposition of this fact).

To extend this algorithm to an infinite pool of experts and an unbounded time horizon, we employ a version of the *doubling trick*. We divide all rounds into disjoint blocks of increasing length, where $k$-th block consists of $2^k$ rounds. In each block, we run the initial algorithm from scratch (i.e., initializing the weights uniformly), with $T = 2^k$ and the first $k$ experts, adjusting the parameter $\eta$ as required for the bound.

Now, consider an expert $e_m$. There are only finitely many rounds in which this expert is not taken into account. In these rounds, in principle, it can happen that the expected regret wrt $e_m$ grows linearly. However, starting from the round $l > \sum_{0 < i < m} 2^k$, $e_m$ is added to the active pool of experts, and the expected regret bound with respect to this expert applies. For $k > m$, the expected regret with respect to $e_m$, in the $k$-th block of rounds, increases by at most $\sqrt{\frac{k}{2} \ln 2^k} \leq \frac{k}{\sqrt{2}}$. In particular, the total expected regret for $\sum_{i=1}^{k-1} 2^i < t \leq \sum_{i=1}^{k} 2^i$ is bounded by $O(1) + \sum_{i=1}^{k} \frac{i}{\sqrt{2}}$, which suffices for sublinearity.

The above expectation is taken with respect to the product measure defined by the probability distributions over the experts considered in each round. We briefly argue how to implement this procedure as an almost total randomized algorithm, with the bound on expected regret holding when expectation is taken with respect to the uniform measure (from which the random advice is sampled).

Given an infinite random advice $x$, we split it into an infinite number of columns $x_1, x_2, \ldots$. Effectively, these columns form an infinite sequence of mutually independent random advices, each sampled from the uniform measure. We will use $x_k$, the $k$-th column of $x$, to decide what to do in the $k$-th round of prediction.

In each round, our prediction procedure needs to sample one expert from a finitely supported probability distribution. The probability of sampling a given expert is a computable real number $r$, meaning that there is a computable procedure, which given $n$, outputs an approximation $q$ such that $|q - r| < 2^{-n}$. In particular, this implies that there exists an almost total measure-preserving mapping from the uniform measure on infinite binary sequences into the set of experts (see, e.g., Theorem 3.1 in Zvonkin and Levin, 1970). We apply this mapping to the $x_k$ to obtain the chosen expert. Since this mapping is finitely-valued, it is enough to read a finite prefix of $x$ to know which expert is chosen.

More precisely, we interpret $x_k$ as a real number in the interval $[0, 1]$ in the usual way and use the correspondence between the uniform measure and the Lebesque measure on that interval. The mapping between infinite binary sequences and real numbers is not unique, but it becomes unique if we omit the rational numbers (which have more than one representation as binary sequences) which form a set of measure zero. Finally, we partition $[0, 1]$ into disjoint subintervals, where each subinterval corresponds to one expert and its length is equal to the probability of sampling that expert. It suffices to read a finite prefix of $x_k$, to know to which of these subintervals $x_k$ is mapped to, except in the case when the real number corresponding to $x_k$ lies on the boundary of the subinterval. There are only finitely many such boundaries, and they form a set of measure zero.

## C   Full proof of Theorem 6

We construct $\mathcal{A}$ as stated in the theorem. The construction proceeds computably in the number of timesteps $s = 0, 1, \ldots$. At each timestep, we have a finite set $A_s$ of programs, each computing an infinite binary sequence. We guarantee that $A_s \subseteq A_{s+1}$ for all $s \in \mathbb{N}$, and define $A = \bigcup_{s=0}^{\infty} A_s$. The class $\mathcal{A}$ is then defined as $\mathcal{A} = \{\varphi_e : e \in A\}$.

The requirements $R_e$ are formulated in Section 4.7. Below, we describe the strategy for $R_e$ more formally. During the construction, the strategy maintains a variable $\gamma_e$ that stores a binary word. At timestep 0, we set $\gamma_e = 0^e 1$ and $\mathcal{A}_0 = \emptyset$. The strategy may be in one of two complementary stages: *active* or *inactive*. Initially, all strategies are *active*.

We use the notation $\varphi_{e,s}(\ldots)$ to denote the execution of at most $s$ instructions of program $e$ on a given input. The program may or may not halt (and return a value) within the time bound $s$. A halting computation (within the time bound) is denoted by $\varphi_{e,s}(\ldots) \downarrow$, and a non-halting one by $\varphi_{e,s}(\ldots) \uparrow$.

$R_e$**-strategy at timestep** $s > 0$**.**

    (1) If $s = e$, set $\mathcal{A}_s := \mathcal{A}_s \cup \{\gamma_e 0^\infty\})$.

    (2) If $s > e$ & (the strategy is *inactive* or $\varphi_{e,s}(\gamma_e, |\gamma_e|) \uparrow$), we do nothing.

    (3) If $s > e$ and the strategy is *active*, and $\varphi_{e,s}(\gamma_e, |\gamma_e|) \downarrow$, then let $b := \varphi_{e,s}(\gamma_e, |\gamma_e|)$.

        (a) If $b = 1$, the $R_e$-strategy changes its state to *invactive*.

        (b) If $b = 0$, set $\mathcal{A}_s := \mathcal{A}_s \cup \{\gamma_e 10^\infty\}$ and $\gamma_e := \gamma_e 1$.

This ends the description of the strategy.

**Construction.**   At each timestep $s = 0, 1, \ldots$, we first set $\mathcal{A}_s := \mathcal{A}_{s-1}$ and then we run the $R_e$-strategies for all $e \leq s$.

The verification is split into lemmas.

**Lemma 6.** $\mathcal{A}$ *is a RER class.*

*Proof.* Given a binary string $\alpha$, $x \in \mathbb{N}$, and $i \in \{0, 1\}$, let $g_i(\alpha, x)$ output $\alpha(x)$ for $x < |\alpha|$ and $i$ otherwise. Let $f_i$, $i \in \{0, 1\}$, be a total computable function such that $\varphi_{f_i(\alpha)}(\cdot) = g_i(\alpha, \cdot)$.[4] In the construction, the operations $\mathcal{A}_s := \mathcal{A}_s \cup \{\gamma_e 0^\infty\}$ and $\mathcal{A}_s := \mathcal{A}_s \cup \{\gamma_e 10^\infty\}$ really mean $A_s := A_s \cup \{f_0(\gamma_e)\}$ and $A_s := A_s \cup \{f_0(\gamma_e 1)\}$, respectively. The set $A = \bigcup_{s=0}^\infty A_s$ is computably enumerable because it can be defined as follows: $n \in A$ if and only if there exists a timestep $s$ at which $n \in A_s$. Clearly, the predicate of two number variables $n, s$ defined by $n \in A_s$ is computable. It follows that $A$ is computably enumerable.[5] Since we define $\mathcal{A} = \{\varphi_e : e \in A\}$, $\mathcal{A}$ is $RER$. $\quad\square$

**Lemma 7.** *$\mathcal{A}$ is computably univerally online learnable.*

*Proof.* We define a $RER$ class $\mathcal{Z}$ such that $\overline{\mathcal{A}} \subseteq \mathcal{Z}$, and apply Theorem 4.

Let $Z = \{f_0(0^e 1^j) : e, j \in \mathbb{N}\} \cup \{f_1(0^e 1^j) : e, j \in \mathbb{N}\}$, where $f_0, f_1$ are as in the proof of Lemma 6, and define $\mathcal{Z} = \{\varphi_e : e \in Z\}$. By the projection theorem (see the proof of previous lemma), $Z$ is computably enumerable because it can be defined as follows: $n \in Z$ iff there exist $e, j$ such that $n = f_0(0^e 1^j)$ or $n = f_1(0^e 1^j)$. It follows that $\mathcal{Z}$ is $RER$.

It remains to show that $\overline{\mathcal{A}} \subseteq \mathcal{Z}$. In Appendix A we observed that $\overline{\mathcal{A}} = \mathcal{A} \cup D(\mathcal{A})$.

First, we prove $\mathcal{A} \subseteq \mathcal{Z}$. If $e'$ is enumerated into $A$, then $e' = f_0(\gamma 1)$, for some $e$, with $\gamma$ of the form $0^e 1^j$, $e > 0$, $j \in N$. It follows that $A \subseteq Z$, so $\mathcal{A} \subseteq \mathcal{Z}$.

Finally, we prove $D(\mathcal{A}) \subseteq \mathcal{Z}$. Let $E = \{0^\infty\} \cup \{0^e 1^\infty : e \in \mathbb{N}\}$. Clearly, $E \subseteq \mathcal{Z}$. Let $\mathcal{A}' = \{0^e 1^j 0^\infty : e, j > 0\}$. $\mathcal{A} \subseteq \mathcal{A}'$ so $D(\mathcal{A}) \subseteq D(\mathcal{A}')$. It is enough to show $D(\mathcal{A}') \subseteq E$. For a contradiction, let $X \in D(\mathcal{A}') \setminus E$. By the properties of $D(\cdot)$:

$$\text{For all } n, \text{ there are infinitely many elements of } \mathcal{A}' \text{ starting with } X \upharpoonright n. \tag{3}$$

$X$ starts with $0$ because every element of $\mathcal{A}'$ starts with $0$. Since $X \neq 0^\infty$, $X$ starts with $0^k 1$ for some $k > 0$. Since $X \neq 0^k 1^\infty$, $X$ starts with $0^k 1^j 0$, for some $j > 0$. But $\mathcal{A}' \cap [0^k 1^j 0] = \{0^k 1^j 0^\infty\}$ which is a finite set. This contradicts (3). $\quad\square$

**Lemma 8.** *No computable online learner for $\mathcal{A}$ is proper.*

*Proof.* We show that every requirement $R_e$ is satisfied. Assume that $\varphi_e$ is a learner and that $\varphi_e$ is proper with respect to $\mathcal{A}$. The are three cases to consider.

1. The value of $\gamma_e$ becomes $0^e 1^j$, $j > 0$, and $\varphi_e(0^e 1^j, e + j) \uparrow$, that is the program $e$ on input $(0^e 1^j, e + j)$ never halts, which causes the $R_e$-strategy to execute step (2) indefinitely thereafter.

   We argue that this case is not possible. Suppose the contrary. Note that since $\gamma_e$ has become $0^e 1^j$, it must have gone through the previous values $0^e 1^k$, for $1 \leq k < j$. Therefore, for each previous value $0^e 1^k$ of $\gamma_e$, the $R_e$-strategy has executed step (b) at some point (otherwise, it would not have reach the current value $0^e 1^j$), and therefore $0^e 1^{k+1} 0^\infty$ was enumerated into $\mathcal{A}$, for $1 \leq k < j$. So $0^e 1^j 0^\infty$ is in $\mathcal{A}$. It follows that $0^e 1^j$ is $\mathcal{A}$-realizable. Since $\varphi_e$ is proper with respect to $\mathcal{A}$, $\varphi_e(0^e 1^j, e + j) \downarrow$, a contradiction.

   It follows that whatever value $\gamma_e$ assumes, we encounter a stage $s$ at which $\varphi_{e,s}(\gamma_e, |\gamma_e|) \downarrow$ and thus we execute step (3).

2. The value of $\gamma_e$ becomes $0^e 1^j$, $j > 0$ and $\varphi_e(0^e 1^j, e + j) = 1$.

   We argue that this case is also not possible. Assume otherwise. By the reasoning above, $0^e 1^k 0^\infty \in \mathcal{A}$ for $1 \leq k \leq j$. Since $\varphi_e(0^e 1^j, e + j) = 1$, we finally execute step (a) while $\gamma_e = 0^e 1^j$, which makes the $R_e$-strategy *inactive*. Afterwards, the $R_e$-strategy always exectues step (2) due to it being *inactive*. Hence, we will never enumerate into $\mathcal{A}$ any hypothesis starting with $0^e 1^{j+1}$. But then $\varphi_e(0^e 1^j, \cdot)$ is not $\mathcal{A}$-realizable, because for an

---

[4]It is computer science folklore that, given a program and an input to it, one can effectively transform it into another program that behaves exactly like the original on that input (see, e.g. Rogers, 1967, Theorem 1-V).

[5]According to the standard characterization of computably enumerable sets, $B$ is computably enumerable if and only if there exists a computable relation $R(x, y)$ such that $x \in B \Leftrightarrow \exists y R(x, y)$. Formal details can be found in Rogers (1967, Corollary 5-XI).

$\mathcal{A}$-realizable sample $0^e 1^j$ and $x = e + j$, the learner $\varphi_e$ outputs 1, yet no element of $\mathcal{A}$ is consistent with this prediction. Hence $\varphi_e$ is not proper—a contradiction.

From this and the previous case, it follows that:

3. For every value of $\gamma_e$, there is a stage at which the $R_e$-strategy executes step (b).

It follows that $\gamma_e$ assumes, in order, all the possible values $0^e 1^j$ for all $j > 0$. This means that $r = 0^e 1^j 0^\infty \in \mathcal{A}$ for all $j > 0$. It follows that $0^e 1^\infty \in \overline{\mathcal{A}}$ and that $\varphi_e(0^e 1^j, e+j) = 0$, for all $j > 0$. By Lemma 1, the sequence $(x, r(x))_{x=0}^\infty$ is $\mathcal{A}$-realizable. But for every $n \geq e$, $\varphi_e$ makes a mistake on the sample $(x, r(x))_{x=0}^n$ and $x = n + 1$. Hence, $\varphi_e$ does not learn $\mathcal{A}$.

$\square$

# D    A graph-coloring example

By Theorem 1, there exist $RER$ (and even $DR$) classes that are universally online learnable but have no computable learner. The proof of this result relies on the existence of computable trees of a certain kind. The notion of a computable tree is defined in Section 4.2. For brevity, and following standard usage in the literature, the set $[T]$ of all infinite paths through a computable tree $T$ is called a $\Pi_1^0$ class.

Although the class of hypotheses from Theorem 1 may appear artificial, the existing literature on applications of $\Pi_1^0$ classes allows us to identify more natural examples. Below, we describe one such example—the problem of learning colorings of a computable graph.

Imagine that a graph is given with a fixed but unknown coloring, and we attempt to learn this coloring. Graph coloring can represent an existing allocation or strategy—such as frequency assignments in wireless networks or conflict-free strategy choices in game-theoretic settings. This learning scenario represents various reverse-engineering problems, in which one seeks to infer the internal logic or policy of a system from its observable outcomes.

Before going into the specifics of our graph-coloring example, let us describe the key idea behind such applications. Given a computably presented problem, the corresponding set of solutions can be represented as a $\Pi_1^0$ class $[P]$. The correspondence between the elements of $[P]$ and the solutions to the problem is typically established via a natural coding of the latter by infinite sequences—paths through a computable tree $P$. As a result, coding-invariant properties that hold for all (some) $\Pi_1^0$ classes automatically extend to the corresponding sets of solutions. For a comprehensive survey of similar applications of $\Pi_1^0$ classes in logic, combinatorics, game theory, and analysis, see Cenzer and Remmel (1998).

A computable graph $G = (V, E)$ consists of a computable set $V \subseteq \mathbb{N}$ of vertices and a computable edge relation $E \subseteq V \times V$. We consider only undirected graphs. A $k$-coloring of $G$ is a map $g$ from $V$ to $\{1, 2, \ldots, k\}$ such that $g(u) \neq g(v)$ for any vertices $u, v$ with $E(u, v)$.

**Learning $k$-colorings of a computable graph.**    Let $G = (V, E)$ be a $k$-colorable computable graph, and let $\mathcal{H}_G$ be some set of $k$-colorings of $G$. In each round, Adversary gives a vertex $v \in V$, Learner outputs a color $\hat{y} \in \{1, 2, \ldots, k\}$, and then Adversary reveals the "true" color $y \in \{1, 2, \ldots, k\}$.

Below we present an example of the above learning problem: a computable graph $G$ and a set $\mathcal{H}_G$ of $k$-colorings of $G$, such that $\mathcal{H}_G$ is universally online learnable but cannot be learned by any computable learner. This example is corollary of the following result by Remmel (1986).

**Theorem 7.** *There exists a $k$-colorable computable graph $G$ with a decidable $k$-coloring problem and a $k$-coloring $g$ such that $g$ is the unique uncomputable coloring of $G$.*

A graph $G$ has a *decidable $k$-coloring problem* if there is a program that decides, for any $k$-coloring $f$ of a finite subgraph $F$ of $G$, whether $f$ is extendable to a $k$-coloring $g$ of $G$.

Let $v_0, v_1, \ldots$ be a 1-1 enumeration of $V$ and let $l \in \mathbb{N} \cup \{\omega\}$ be the length of this enumeration. A $k$-coloring of $G$ may be represented as a sequence $X \in \{1, 2, \ldots, k\}^l$ such that $X(i) \neq X(j)$ whenever $E(v_i, v_j)$. We let $G{\upharpoonright}n$ denote the induced subgraph of $G$ with vertex set $\{v_0, v_1, \ldots, v_{n-1}\}$.

**Lemma 9.** *Suppose $G$ is a computable $k$-colorable graph with a decidable $k$-coloring problem. Then there exists a program computing a function $c(f, n)$ with the following property: for any $n \in \mathbb{N}$ and any $f : \{0, 1, \dots, n-1\} \to \{1, 2, \dots, k\}$, if $f$ is a $k$-coloring of $G{\restriction}n$ and $f$ is extendable to a $k$-coloring $g$ of $G$, then $c(f, \cdot)$ is a $k$-coloring of $G$ extending $f$.*

The proof of the above lemma is straightforward.

We now define our example of a graph-coloring problem. Let $G_0 = (V_0, E_0)$ be a graph from Theorem 7.

**Definition 6.** *Let the class $\mathcal{H}_{G_0}$ consist precisely of those functions $c(f, \cdot)$ for which there exists $n \in \mathbb{N}$ such that $f$ represents a $k$-coloring of $G_0{\restriction}n$ that is extendable to a $k$-coloring of $G_0$.*

It can be easily shown that $\mathcal{H}_{G_0}$ is $RER$.

**Lemma 10.** $\overline{\mathcal{H}_{G_0}}$ *represents the set of all $k$-colorings of $G_0$.*

*Proof.* If $g \in \overline{\mathcal{H}_{G_0}}$, then every initial segment $g{\restriction}n$ represents a $k$-coloring of $G_0{\restriction}n$. If $g$ were not a $k$-coloring of $G_0$, then there would exist $i, j$ such that $g(i) = g(j)$ and $E_0(v_i, v_j)$; but then any sufficiently long initial segment of $g$ would not represent a $k$-coloring of the corresponding subgraph of $G_0$.

For the other direction, if $g$ is a $k$-coloring of $G_0$, then every initial segment $g{\restriction}n$ is a $k$-coloring of $G_0{\restriction}n$ that can be extended to a $k$-coloring of $G_0$. Therefore, $c(g \restriction n, \cdot) \in \mathcal{H}_{G_0}$, which shows that $g \in \overline{\mathcal{H}_{G_0}}$. $\qquad\square$

It follows immediately from Theorem 7 that the class $\overline{\mathcal{H}_{G_0}}$ is countable. By Lemma 4, $\mathcal{H}_{G_0}$ is not computably universally online learnable because $\overline{\mathcal{H}_{G_0}}$ contains an uncomputable member. It remains to observe that $\mathcal{H}_{G_0}$ is universally online learnable. Since $\overline{\mathcal{H}_{G_0}}$ is countable, arrange its elements into a sequence $h_0, h_1, \dots$. A universal online learner for $\mathcal{H}_{G_0}$ works as follows: on input $(S, x)$, where $S$ is a sample obtained in previous stages of the game, output $h_i(x)$, where $i$ is the least $j$ such that $h_j$ is consistent with $S$. By Lemma 2, any realizable sequence arising from Adversary's moves is consistent with some element of $\overline{\mathcal{H}_{G_0}}$, so the described learning strategy will eventually stop making mistakes.

We note that the previous paragraph requires reproving the lemmas from Section 4.1 for the topological space $\{1, 2, \dots, k\}^\omega$ which is the product space of the discrete space $\{1, 2, \dots, k\}$. The Cantor space $2^\omega$ can be obtained in a similar way as a countable product of the discrete space $\{0, 1\}$. In general, the proofs of lemmas from Section 4.1 depend on the properties of the underlying topological space, such as compactness and metrizability. It is known that a countable product of compact spaces is compact with respect to the product topology (Tychonoff's theorem). Moreover, there exists a metric such that the topology on $\{1, 2, \dots, k\}^\omega$ induced by that metric coincides with the product topology (see, e.g., Engelking, 1989, Theorem 4.2.2., p. 259).

