# OpenReview forum: "Computable universal online learning"
_NeurIPS.cc/2025/Conference — NeurIPS 2025 poster_

### Official Review · Reviewer_A9r9 · 2025-06-23

**Clarity:** 3
**Significance:** 2
**Originality:** 3
**Rating:** 3
**Confidence:** 4

**Summary:**

This submission studies universal learning from a computability perspective. Universal Learning is a theoretical model of online binary classification where an adaptive Adversary can change the target concept (rather than fixing it ahead of time). Prior work (e.g., Bousquet et al. 2021) characterized universal learnability. This paper adds the constraint that the Learner must be implementable as a computer program. In this case, no characterization was known. The submission also studies proper computable universal learning.

The submission shows:
- Universal learnability does not imply computable universal learnability, even for recursively enumerably representable (RER) classes.
- A characterization for when a hypothesis class is computably learnable in the agnostic setting (here, the Adversary can choose any sequence, not just one belonging to the class).
- A characterization for when proper computable learning is possible.
- Separations that indicate that some classes are computably universally learnable but not computably agnostically learnable, and some RER classes are computably universally learnable but not by any computable proper learner.

**Questions:**

1. How does randomized sampling on LLMs fit into example 3, if at all?
2. Proposition 1 shows that even DR (decidably representable) classes might not be computably universally learnable. Is there some intuition on what aspects of a DR class leads to the noncomputability of a universal learner?

**Ethical Concerns:**

["NO or VERY MINOR ethics concerns only"]

**Final Justification:**

I recommend a score of 3 (Borderline Reject).

While the paper is technically solid, its central contributions have limited significance for the NeurIPS audience (in my opinion). The main results link properties of hypothesis classes (like RER and DR), which relate to their "describability," to a very weak, binary notion of learnability (whether an algorithm is computable or not). This is a step away from more informative and fine-grained efficiency questions that are central to machine learning.

The proof techniques, such as diagonalization-style "evil sequence" arguments from TCS, are applied here to establish results whose practical implications for machine learning remain unclear.

**Limitations:**

yes

**Quality:**

3

**Strengths And Weaknesses:**

**Strengths**
- I find the paper to be written pretty clearly, even as someone who has not studied universal learning in depth.
- The paper makes progress on and in some cases (e.g., characterizations) fills what stands as a gap in the literature on the topic.
- The paper's topic is fundamental: what are the limits of universal learning when we require that the learner be implementable as a program?

**Weaknesses**
- The paper feels much better aligned with theoretical computer science venues such as COLT, STOC/FOCS, and even CCC (computational complexity conference). While the connection to "learning" is clear, in my opinion, the paper doesn't demonstrate why these specific computability results will inspire future research in the typical scope of Neurips (relative to COLT, STOC/FOCS, CCC).
- The paper motivates the question it studies (we should be able to implement algorithms by a program), but it does not sufficiently motivate why answering this kind of question with computability theory is appropriate.
- It is hard to interpret example 3 (LLMs and universal learning) since it is stated "At a high level of abstraction."

**Overall:**
I understand that computability is an established, classical approach to complexity. And I also agree that initial motivation is good, which is that universal learning should be studied with attention to the fact that learning algorithms, for all intents and purposes, should be implemented as a computer program. However, I'm not convinced that characterizing computability (in the sense defined) of universal learning actually answers this question in a practically meaningful way.

---

> ### Author Rebuttal · Authors · 2025-07-30
>
> Thank you for your review!
>
> Regarding computability, it certainly makes sense to study universal learning under stricter complexity bounds—such as primitive recursiveness or polynomial-time computability. However, the most general notion of effectiveness (i.e., computability) provides a natural starting point for analysis. This approach encompasses all practically meaningful learners. Negative results in this setting are especially informative: if no computable learner exists, then there can be no primitive recursive or polynomial-time learner either.
> From a presentation standpoint, computability-theoretic proofs are often cleaner and easier to understand than those involving finer-grained bounds like polynomial-time or primitive recursion. At the same time, NeurIPS has already published papers on related formal frameworks (even without any computational constraints) in the recent past (see for instance Z. Lu "When Is Inductive Inference Possible?", NeurIPS 2024). From this perspective, our paper makes a small step closer towards the practical aspects as compared to pure learning theory papers with no computability.
>
> Ad 1. We can think that randomized sampling in LLMs corresponds to decisions made by the Adversary (each path in the tree of random choices is a different possible adversarial behavior). As usual in adversarial setting, we are interested in the worst-case scenario (what is the worst we can see while doing random sampling) as opposed to average case.
>
> Ad 2.  Yes, there is a short intuition. In our setting the Adversary is adaptive, which means that they can effectively use a hypothesis outside of the DR or RER class $\mathcal{H}$, even if locally at each round the target hypothesis is consistent with some function from $\mathcal{H}$. To be more precise, the Adversary can use any function from the topological closure $\overline \mathcal{H}$.  A DR class of hypotheses $\mathcal H$ may have a non-computable element $h$ in its closure $\overline \mathcal{H}$.  The intuition is that we can approximate a non-computable object by an infinite sequence of computable ones. To take the LLMs example, any finite sequence of tokens produced by an LLM is computable (even under randomized sampling), but if we let the token generation run indefinitely, we can obtain a sequence that is no longer computable (hence, outside of computable learner's grasp).

---

> ### Comment · Reviewer_A9r9 · 2025-08-02
>
> Thank you for your response.
>
> While I understand your point about the negative results, in characterizations there also exists the "positive" direction, in which case computability is a low bar.
>
> I'm willing to consider the argument on precedence of computability work (or work without computational constraints) in NeurIPS. However, I had not used the scope of the paper to penalize the submission's rating significantly, so I won't change my score.

---

> > ### Author Response · Authors · 2025-08-04
> >
> > Thank you for your answer,
> > In your initial review, you have mentioned three weaknesses:
> > 1. Better alignment with other conferences (e.g., COLT) and no explanation why our results may inspire future research in the typical scope of NeurIPS.
> > 2. Use of computability as a metric of effectiveness.
> > 3. Problems with interpreting Example 3.
> >
> > Example 3 has been further explained in the rebuttal. The example does not play a central role in the paper’s overall argument or contributions. Regarding points 1 and 2, we understood that the reviewer downplays their significance in the final evaluation, stating: “I had not used the scope of the paper to penalize the submission's rating significantly, so I won't change my score.”
> >
> > Therefore, we would like to kindly ask the Reviewer to further clarify: which aspect of the submission was the main cause of the negative rating? Thank you again for your responses!

---

> > > ### Comment · Reviewer_A9r9 · 2025-08-05
> > >
> > > > Regarding points 1 and 2, we understood that the reviewer downplays their significance in the final evaluation, stating: “I had not used the scope of the paper to penalize the submission's rating significantly, so I won't change my score.”
> > >
> > > I did not use my opinion that the paper is better suited to several top TCS venues such as COLT/STOC/FOCS/CCC significantly to influence my score.
> > >
> > > To derive my score, I did use my view that a very coarse notion of algorithmic learnability such as computability leaves a lot on the table in terms of the "positive" direction in the characterizations, e.g., if complement of H is RER, then H is properly computability universally learnable. While it is technically correct that computability corresponds to ability to implement on a computer, this doesn't correspond to ability to implement as a program in any practically meaningful sense, especially in the context of machine learning.
> > >
> > > Overall, I believe the paper is technically solid, where reasons to reject outweigh reasons to accept.

---

### Official Review · Reviewer_b4op · 2025-06-28

**Clarity:** 4
**Significance:** 3
**Originality:** 3
**Rating:** 5
**Confidence:** 4

**Summary:**

This work investigates when learners that achieve objectives such as universal learning (i.e., making only finitely many mistakes in the limit) can be made computable in the online binary classification setting. Specifically, it asks: for which hypothesis classes $\mathcal{H}$ does there exist a computable online learner that makes only finitely many mistakes on every realizable example sequence $(z_t) = (x_t, y_t)$ such that for any $t \in \mathbb{N}$, there exists a hypothesis $h \in \mathcal{H}$ such that $y_s = h(x_s)$ for all $s \le t$? Note that this definition allows the realizing hypothesis $h_t$ to vary with the finite horizon $t$, a condition the authors call local consistency of the labels with $\mathcal{H}$.

Previous work has provided a characterization of universally learnable hypothesis classes. However, the question of whether learners achieving the finite-mistakes objective can be made computable has remained unaddressed until now. In this sense, the present work fills a fundamental gap by focusing on the algorithmic implementability of the online learner, emphasizing that a learning "procedure" must be an algorithm (in the Turing sense), not merely an explicit mathematical construction.

The main contributions are several. First, the authors show that even if $\mathcal{H}$ is recursively enumerably representable (RER), this alone does not guarantee the existence of a computable universal learner. Undeterred by the failure of this natural sufficient condition, the authors develop a deeper characterization, which falls short of their initial goal but still yields significant and interesting results.

It is shown that $\mathcal{H}$ admits a *total* computable universal learner if and only if the topological closure of $\mathcal{H}$ is contained in an RER set. Note that “computable” alone refers to partial computability, meaning the learner may be undefined or fail to halt on some inputs in its domain. Hence, the existence of a total computable learner is a strictly stronger assumption than merely requiring computable universal learning. Moreover, the aforementioned conditions are equivalent to the existence of a computable agnostic learner for $\mathcal{H}$, where a computable agnostic learner is a randomized, almost-total learner that achieves sublinear regret relative to $\mathcal{H}$ on any example sequence, not just those realizable by $\mathcal{H}$. A complete characterization of computable universal learning, which was perhaps the original inspiration for this work, remains an open problem.

**Questions:**

1. **Are the authors assuming that each hypothesis in the given class is a total computable function?** Identifying $h$ with an infinite binary sequence in Cantor space appears to require that every entry of the sequence is defined. Moreover, in parts of the proof where the consistency of $h$ with the current finite history of examples is checked, it seems crucial to ensure that a defined (halted) value for $h(x_s)$ exists for all $s \le t$.

2. **Related to the above issue, Lemma 5 seems to require that every expert is a total function.** If an active expert's prediction is undefined on feature x_t, then we cannot compute the exponential weights update for time t+1.

3. **Proof of Theorem 5, (3) => (2) is unclear. In particular, in lines 310-314 and 324-327.** I do not see the necessity of introducing a new probability measure $P'$. Given the $[0,1]$-valued forecasts produced by the multiplicative weights update (MWU) algorithm, which the authors denote as the randomized no-regret algorithm $L$, the goal is to construct a deterministic online learner $L'$. The authors achieve this by simulating all random advice sequences at round $t$ until all but a $1/(k+1)^2$ fraction have halted, and taking a majority decision. However, the majority decision yields a deterministic value with no residual randomness because the random advice sequences are simulated according to a fixed (potentially dovetailed) enumeration to ensure that $L'$ is deterministic. Therefore, the introduction of the probability measure $P'$ seems unnecessary and is confusing to me.\
\
The quantity of interest seems to be the gap between the expected 0/1 loss of the randomized prediction $\hat{y}_t$, output by $L$, and the majority decision $\bar{y}_t$, a deterministic quantity once the enumeration of random advice sequences is fixed, computed from all but a $\delta_k$ fraction of the random advice sequences on which $L$ halts. Using the inequality $(\bar{y}_t \neq y_t) \le (\bar{y}_t \neq \hat{y}_t) + (\hat{y}_t \neq y_t)$, where $y_t$ is the observed label, and then taking the expectation over $\hat{y}_t$, appears sufficient to analyze the regret gap.

**Ethical Concerns:**

["NO or VERY MINOR ethics concerns only"]

**Final Justification:**

Rating remains the same. Reasons are provided in my response to the authors' rebuttal.

**Limitations:**

yes

**Quality:**

3

**Strengths And Weaknesses:**

### Strengths

1. **Addresses a significant and often overlooked gap in the field.** The majority of works in learning theory, aside from a recent line of remarkable contributions, leave the issue of computability entirely unaddressed. These works often incorrectly refer to explicit mathematical procedures as “algorithms,” a term with a precise technical meaning established by Gödel, Church, and Turing, without formally considering their computability. I therefore find it greatly satisfying that this work rigorously tackles the computability question in the context of online learning.
2. **Proof techniques are particularly illustrative.** The authors identify hypotheses on the input domain $\mathbb{N}$ with elements of Cantor space, which is a complete metric space (w.r.t. the Baire metric on infinite sequences) and hence a well-defined topological space. This identification enables a clean mathematical formulation and facilitates the use of well-established tools from analysis.
3. **Excellent exposition.** I believe this level of clarity is a significant contribution on its own, given the technical nature of the notions and proof techniques involved in computability. Minor clarifications could further improve it, especially since computability is highly sensitive to precise specification of the objects involved (see **Questions** for details).

### Weaknesses
1. **Setup is clean but lacks a connection to standard hypothesis classes.** For example, which reasonable subsets of linear threshold functions (LTFs) over $\mathbb{S}^{d-1}$ are RER? This gap could leave the exposition feeling incomplete. The authors’ statement,
> ... in this paper we limit our attention to universal learning of hypotheses defined on $\mathbb{N}$. This might seem limiting but this choice is guided by practical considerations. Indeed, all practical machine learning happens on computers, which inherently work with countable domains, such as images or binary strings. From the point of view of computability, learnability on natural numbers is sufficiently general to model any domain relevant to practical applications.
>
reads more like a deflection than a rigorous argument, as it does not directly address how standard hypothesis classes like LTFs can be formalized within this framework. While it is reasonable to view focusing on $\mathbb{N}$ as a necessary first step, it should also be recognized that many standard hypothesis classes, particularly those defined over continuous domains with real-valued parameters like threshold functions on [0,1], are technically not RER and thus fall outside the current formalism.\
Extending this computability framework to encompass more standard hypothesis classes, such as LTFs defined on unit spheres in $\mathbb{R}^d$, remains an important connection to establish and will require careful, meticulous formalization, even if it does not involve fundamentally new ideas.

2. Some aspects of the setup and proofs are unclear, as detailed in the **Questions** section.

---

> ### Author Rebuttal · Authors · 2025-07-24
>
> Thank you a lot for such a positive review!
>
> Ad. 1 and 2. You are right in both cases. We assume that each hypothesis is a total computable function and indeed, we should have stated in Lemma 5 that each expert is total as well. Thank you for noticing this.
>
> Ad 3. In our argument, the no-regret randomized learner is only assumed to be almost total. This means that the set of random advices on which the algorithm halts is of measure 1 but still it is possible that the algorithm does not halt on some random advice. In such case we cannot always derandomize the algorithm into a computable majority vote. If the algorithm is defined everywhere, on every random advice, we can just enumerate all the random oracles until at some point the computations will stop on exactly measure 1 set of random advices. But if the algorithm is only almost everywhere total, then enumerating random oracles only guarantees that we will keep seeing larger and larger measure, approximating $1$ but possibly never reaching it. Roughly speaking, what can happen is that both label 1 and label 0 will keep having approximately 1/2 of all the measure we have enumerated so far and we will never know what is the value of majority vote. For instance, it could happen that our randomized algorithms halts on random advices of form 01..10 (outputting 0) and 1..10 (outputting 1), in which case we have a tie. But seeing always only a finite set of oracles we cannot distinguish this from the case when for some advice of form 01....1 the output is 1, in which case the majority vote should give 1.
>
> That's why we need this small trick, in which we say that we will care only about the measure that is very close to $1$ (this is our $P'$). On this restricted set of random advices the majority is computable and we then simply note that any additional mistakes we have got this way can be neglected, since we are only interested in the asymptotic property of sublinear regret.

---

> > ### Comment · Reviewer_b4op · 2025-08-04
> >
> > I thank the authors for their clarification. My positive assessment of the paper remains unchanged.
> >
> > I would like to take this opportunity to support the authors and gently push back on some points raised by other reviewers, as I believe this work addresses a valuable and underexplored direction.
> >
> > **On the importance of computability.** Computability is, of course, too coarse of a notion to capture practical efficiency. It's certainly not sufficient, but it is *the* minimal requirement for something to formally qualify as an "algorithm". As such, it serves as a necessary first step toward understanding what kinds of procedures are even implementable in principle, let alone practical.
> >
> > This paper asks a fundamental question: Do results from online learning still hold once we take computability into account? Is the common practice of ignoring computability entirely harmless from the standpoint of mathematical validity? The answer is far from clear, and this work demonstrates that subtle counterexamples can and do arise. It suggests that we should tread more carefully.
> >
> > **On the connection to Bousquet et al. (2021).** Two reviewers have raised concerns about the paper's reference to Bousquet et al., particularly regarding the batch vs sequential distinction. While I have not verified the details myself, even if the stated connection is erroneous, I do not believe it undermines the core contributions of the paper. Its value lies in the rigorous treatment of computability in the context of online learning, which stands on its own.

---

### Official Review · Reviewer_iVFU · 2025-06-30

**Clarity:** 2
**Significance:** 3
**Originality:** 2
**Rating:** 4
**Confidence:** 3

**Summary:**

Recently, it has been shown that PAC learning might be undecidable, and that learnability cannot be determined within ZFC. In the model of computational PAC learning, where algorithms must computably output predictors, the main idea is that the VC dimension no longer characterizes learnability. Instead, a "computational" version of the VC dimension, known as the effective VC dimension, takes its place. Similar results have been shown for online learning and the Littlestone dimension.

This paper studies the computational version of universal learning, a model introduced by Bousquet et al., STOC 2021. This model lies between classical universal consistency and PAC learning. Unlike PAC learning, the sample complexity in this model depends on unspecified constants that themselves depend on the (fixed but unknown) data distribution.

My main concern is that the learning model described in the paper does not actually correspond to universal learning, see below.

**Questions:**

My main concern is about the definitions of the learning model, see above.

**Ethical Concerns:**

["NO or VERY MINOR ethics concerns only"]

**Final Justification:**

POST-REBUTTAL:
The authors clarified the exact setting they are studying and suggested simple modifications to improve the clarity of the presentation.
Therefore, I am updating my score.

**Limitations:**

Limitations are addressed.

**Quality:**

3

**Strengths And Weaknesses:**

I believe there is considerable confusion in the paper regarding the definition and scope of the universal learning setting as introduced by Bousquet et al. (STOC '21). In particular, the authors refer to this model as an online learning setting, which is not accurate. For example:

Abstract: "We address this question for universal learning, a generalist theoretical model of online binary classification, recently characterized by Bousquet et al. (STOC´21)."
The model proposed by Bousquet et al. is not an online learning model. Although online learning techniques are used within the algorithm, the setting itself is batch, not sequential.

Lines 30–31: "This setting aligns with real-world scenarios where uniform bounds are not always available and where data distributions may shift."
This is misleading. The distribution in this setting is fixed in advance and does not shift. The key feature is that the sample complexity may depend on the distribution (via unknown constants), but the distribution itself does not change with the sample size.

 The paper claims to study “agnostic computable universal learning,” but the characterization of even the simpler model of agnostic universal learning is still unresolved. Without a full understanding of the base model, it is unclear how one can meaningfully study a more complex, computability-based variant.

Definition 1: This is neither the standard definition of universal learning nor a rigorous formalization.

Additional Comments:

Line 20 (Introduction): It is unclear what relevance multi-armed bandits have to the discussion in this paper.

Line 28: "Adversary fixes one true function H in advance."  There seems to be a typo here.

Definition 2: The definition of the agnostic setting appears to be incorrect. It is not clear what learning model is being formalized here.

Finally, I must note that some sections may have been generated or heavily by an LLM, especially the introduction.

---

> ### Author Rebuttal · Authors · 2025-07-24
>
> Let us elaborate on the relation between our definitions and the work of Bousquet et al 2021. In the full version of their paper (posted on arxiv) there are at least two learning settings that are called universal learning.
> You are right that Bousquet et al. 2021 was primarily concerned with studying learning rates under probabilistic sampling. But a crucial technical contribution of their paper - announced in the main paper and developed in the full version - is the notion of ordinal Littlestone dimension and showing that ordinal Littlestone dimension characterizes online learning without uniform mistake bound under Adversarial sampling (compare this to the line 28 of our paper). Bousquet et al 2021 mention this setting on page 535 (the paragraph below Definition 1.7) of the conference paper and refer to the full arxiv version for details, where they call it the "universal variant of online learning" (Section 3.1 on Page 15 of the full arxiv version of their paper). This name was further used by Hanneke et al (ALT 2025). We understand that this may seem somehow confusing and personally, we would prefer to call this setting “non-uniform online learning”. However, in the light of this setting being called “universal learning” and “universal online learning” by other authors (Bousquet et al. 2021, Hanneke et al 2025), we decided to follow the established terminology.
>
> Indeed, our Definition 1 simply repeats the earlier definition of “universally online learnable” as introduced by Bousquet et al (page 15 of the full paper) and then generalized to multiclass learning on page 9 of Hanneke et al 2025. To avoid confusion we will extend the definition with additional explanation: We say that $L$ universally online learns $H$ if for every $H$-realizable sequence $(x_t,y_t)_{t>0}$ there exists $m$ such that for all $n>m$ and every sample $S=(x_1,y_1),(x_2,y_2),...$ we get $L(S,x_n)=y_n$. Again, please note that this definition simply follows the work of previous authors, including that of Bousquet et al.
>
> Similarly, our definition of agnostic learning is adapted from the Definition 6 of Hanneke et al 2025. Please note that we reference this paper in the line 151 when we introduce the agnostic setting. We cannot agree that this definition is incorrect or that it is not clear what setting we formalize, as we give a precise reference to the paper from which we took it. So indeed, for the agnostic learning defined as having sublinear expected regret, it is already known that both realizable and agnostic settings are fully characterized by the ordinal Littlestone dimension, as shown by Hanneke et al 2025. We hope that it is now clear that the base model is already fully understood.
>
> We will disambiguate our citation of Bousquet et al. 2021 to clearly point to the right (adversarial online) setting.  We also propose to change “universally learnable” to “universally online learnable” (and similarly “agnostic learning” -> “agnostic universally online learning”) in any occurrence of these notions in the text. This is an easy fix that should help avoid confusion. We will also add a brief comment in the introduction to make it clear that there are two meanings of universal learning in the full version of Bousquet et al. 2021 and that we are concerned with the online one. We humbly note that we have repeated multiple times that we work with an online learning setting.
> Furthermore, we have provided all relevant definitions in the text, which should clear any confusion. We believe that our presentation, together with the suggested improvements, is of sufficient mathematical rigour, similar to that usually encountered in learning theory papers.
>
> The expression “where data distributions may shift.” refers to the fact that in the universal online learning setting the Adversary is adaptive and can change the target function. We will remove this sentence to avoid unnecessary confusion. Similarly, “multi-armed bandit” are mentioned simply as an example of many formal models studied in learning theory.

---

> > ### Comment · Reviewer_iVFU · 2025-08-05
> >
> > Thanks for your response.
> > I think the changes make sense and clearly clarify the setting, the new terminology is also quite helpful for the presentation.
> > Given these updates, I'll raise my score from 2 to 4 (borderline accept).

---

### Official Review · Reviewer_K2dL · 2025-07-02

**Clarity:** 2
**Significance:** 3
**Originality:** 3
**Rating:** 5
**Confidence:** 3

**Summary:**

The authors introduce computability requirements to the framework of universal learning and show:
- That there exists a recursively enumerable class that is universally learnable but not computably universally learnable.
- A characterization of agnostic computable universal learning
- A characterization of a variant that requires algorithms to be proper

**Questions:**

Could you comment on the set-up/my confusion between the universal learning set-up of Bousquet et al. (2021) and the universal vs uniform online learning set-up mentioned in the weakness section? It is a bit difficult to evaluate the rest of the paper without that issue clarified.

Could you give more intuition/details on your definition of proper learning, as it is not really standard?

There are also proper/improper separation results in standard CPAC learning (Delle Rose et al., 2023) -- how do your results compare?

**Ethical Concerns:**

["NO or VERY MINOR ethics concerns only"]

**Final Justification:**

Upped my score 3 -> 5 (see response to reviewers for justification)

**Limitations:**

yes

**Quality:**

2

**Strengths And Weaknesses:**

## Strengths
- Novel set-up: computability has not yet been considered for universal learning, of interest to the learning theory community
- A variety of results are shown, with very few lose ends
- The technical weight of the proofs seems adequate for NeurIPS

## Weaknesses
- There isn't much context/review/explanations about universal learning, which is relatively recent (2021). I would guess most of the NeurIPS readership (even learning theory folks) would not be familiar enough with these concepts for the level of explanations given here. If the authors intend on submitting in a conference like NeurIPS, more explanations should be given on this set-up.
- Likewise, there is a lack of clarity in the beginning of Section 2.1. Most readers will be familiar with online learning, but, at first glance, lines 88-92 do not make it clear what is the difference with the more common set-up of online learning and this one: even in (uniform) online learning the adversary is allowed the flexibility to change the target as long as it is consistent with the history. What differentiates uniform and universal online learning is that, in the former, the number of mistakes has to be uniformly bounded (there exists a learner $\mathcal A$ and integer $d\in\mathbb N$ such that *for any* $h\in \mathcal H$ , $\mathcal A$ makes at most $d$ mistakes) vs in the uniform online learning, the number of mistakes is simply required to be finite (so, there is an algorithm $\mathcal A$ such that for any $h\in\mathcal H$ there exists $d_h\in\mathbb N$ such that $\mathcal A$ makes at most $d_h$ mistakes). This is distinction between set-ups is not clear at all, and would be confusing for most readers who are only/mostly familiar with uniform online learning.
- I am not very familiar with universal learning, but my understanding is that there is a distribution at hand (in a PAC-like set-up, but where the learning rate can also be distribution-dependent), and that the universal online learning results are tools to take back to this main set-up. In particular, the error in universal learning depends on the distribution at hand (as in p.2-3 of Bousquet et al., 2021), and a central concern in universal learning is that of learning rates (not really studied here). Here it seems the paper is mostly concerned about the online version of universal learning? In any case, the authors do not provide any explanations for the departure from the universal learning set-up of Bousquet et al. (2021)
	- An attempt was made for Definition 1 to be readable/intuitive, but this was at the cost of clarity and precision, which is not ideal considering that the whole paper is based on this set up. Please include appropriate mathematical objects ( $\mathcal H$ , $X$ , labels $y_t$ ) and quantifiers. The authors should also reference the work of (Bousquet et al, 2021) in the definition (and the appropriate works in any subsequent definition)
- There are basic definitions missing:
	- For example, a computable learner not only halts on every sample, but also output functions that are themselves total computable (see Definition 8 in Argarwal et al., 2020). Most NeurIPS readers will *not* be familiar with computable learning frameworks, and some more effort should be made in the write-up to introduce these concepts, and also attribute the definitions to the works they originally belong to.
- The authors state many literature results without references:
	- l.127-129: Delle Rose et al., 2025.
	- l.184: not sure where this comes from. Definition 2 should read agnostic online learning I believe.  In any case, universal learning depends on the learning rates, so you have classes that are universally learnable but at arbitrarily slow rates, and thus not PAC learnable (implying not online learnable). So I am not sure this assertion is true -- but it is hard to check without a proper reference.
- Overall, the writing lacks clarity, precision, rigour and the necessary definitions/ideas to get the point through to a NeurIPS audience (including learning theory folks), which has been a big factor in my score. I am a bit confused by the various set-ups and the difference with the Bousquet et al. (2021) paper (which I read briefly to prepare for this review).

## A note on writing:
- Learner vs learner (l.139): it would be better to have different names. Why not simply "strategy"?
- The writing could be improved (e.g., l.169 "In agnostic learning, it only makes sense when the Learner has a strategy defined on all possible inputs, including non-realizable ones", awkward phrasing, l.184 "in standard theory" - of what?, etc., and there are many instances where an article is missing before the noun, e.g., l.13 "We then study agnostic variant of" -> "We then study the agnostic variant of"; l. 25 "a strategy that guarantees uniform bound on mistakes" -> "a strategy that guarantees a uniform bound on mistakes", etc.)
- In Section 4, it would be worth restating the theorems being proved for readability (right now, the reader has to go back and forth)
- I am also generally not a fan of anthropomorphizing learners/ adversaries (even less so with only "he" for all parties). Perhaps the authors would consider changing "he/his to "it/its"
- While not crucial, it would also be helpful to have definitions of DR, RER, closure of $\mathcal H$ , and general computability definitions in proper definition environments for readability.

---

> ### Author Rebuttal · Authors · 2025-07-24
>
> Let us elaborate on the relation between our definitions and the work of Bousquet et al 2021. In the full version of their paper (posted on arxiv) there are at least two learning settings that are called universal learning.
> You are right that Bousquet et al. 2021 was primarily concerned with studying learning rates under probabilistic sampling. But a crucial technical contribution of their paper - announced in the main paper and developed in the full version - is the notion of ordinal Littlestone dimension and showing that ordinal Littlestone dimension characterizes online learning without uniform mistake bound under Adversarial sampling (compare this to the line 28 of our paper). Bousquet et al 2021 mention this setting on page 535 (the paragraph below Definition 1.7) of the conference paper and refer to the full arxiv version for details, where they call it the "universal variant of online learning" (Section 3.1 on Page 15 of the full arxiv version of their paper). This name was further used by Hanneke et al (ALT 2025). We understand that this may seem somehow confusing and personally, we would prefer to call this setting “non-uniform online learning”. However, in the light of this setting being called “universal learning” and “universal online learning” by other authors (Bousquet et al. 2021, Hanneke et al 2025), we decided to follow the established terminology.
>
> Indeed, our Definition 1 simply repeats the earlier definition of “universally online learnable” as introduced by Bousquet et al (page 15 of the full paper) and then generalized to multiclass learning on page 9 of Hanneke et al 2025. To avoid confusion we will rewrite lines 88-92, highlight the main difference between uniform and non-uniform set-ups and extend the definition with additional symbolic explanation: We say that $L$ universally online learns $H$ if for every $H$-realizable sequence $(x_t,y_t)_{t>0}$ there exists $m$ such that for all $n>m$ and every sample $S=(x_1,y_1),(x_2,y_2),...$ we get $L(S,x_n)=y_n$.
>
> Similarly, our definition of agnostic learning is adapted from the Definition 6 of Hanneke et al 2025. Please note that we reference this paper in the line 151 when we introduce the agnostic setting. So indeed, for the agnostic learning defined as having sublinear expected regret, it is already known that both realizable and agnostic settings are fully characterized by the ordinal Littlestone dimension, as shown by Hanneke et al 2025 (Theorem 3 there).
>
> We will disambiguate our citation of Bousquet et al. 2021 to clearly point to the right (adversarial online) setting.  We also propose to change “universally learnable” to “universally online learnable” (and similarly “agnostic learning” -> “agnostic universally online learning”) in any occurrence of these notions in the text. This is an easy fix that should help avoid confusion. We will also add a brief comment in the introduction to make it clear that there are two meanings of universal learning in the full version of Bousquet et al. 2021 and that we are concerned with the online one. We humbly note that we have repeated multiple times that we work with an online learning setting.
> Furthermore, we have provided all relevant definitions in the text, which should clear any confusion. We believe that our presentation, together with the suggested improvements, is of sufficient mathematical rigour, similar to that usually encountered in learning theory papers.
>
> Regarding flexibility of the Adversary. In the uniform case, it doesn’t matter if the Adversary can change the target function. If the Littlestone dimension is finite, then we can achieve uniform mistake bound even if the Adversary is adaptive and if the Littlestone dimension is not finite, Adversary does not need to be adaptive to force $k$ mistakes for arbitrary $k$. Perhaps for this reason, many authors choose to say instead that the target function is fixed in advance (for instance, see the standard textbook by Shai Shalev-Shwartz and Shai Ben-David). In fact, this is how the setting was originally defined in Littlestone 1998. The flexibility only matters when we drop the requirement of uniformity. This is why we highlighted this aspect in our presentation.
>
> Regarding computable learners. In online learning a learner outputs a prediction on a point, not a whole function. The definition 8 from Agarwal et al. 2023 is a definition of computable learner in cPAC setting, not in online learning. Our definition of a learner (see line 140) follows that used by previous authors, both in standard online learning and in its computable version e.g. Hasrati and Ben-David 2023 or Delle Rose et al 2025. A computable learner is just a function $f$ such that $f$ is a learner and $f$ is computable. We use “computable” in the standard sense. The only subtle point in our setting is concerned with learner being a partial or total computable function but this distinction is discussed in detail in the paper.
>
> We will add the reference to the Lines 127-129. This reference is not Delle Rose et al 2025. Please note that the correct references were already given when the claim first appeared in line 50.
>
> Regarding proper learners. First of all, let us say that for the universal online learning, there is no established notion of 'proper learning' yet. In other learning settings, “proper” usually means 'predicting always according to some function from the class'. In the non-uniform online learning, this notion is too strong, since the Adversary can effectively use a function that is not in the class (but rather in its topological closure). Therefore, to be able to say something non-trivial, we decided to define 'proper universal online learning' as 'predicting always according to some realizable function".
> The Reviewer also asked about proper vs improper learning in cPAC setting. This scenario differs in many ways from our case. Proper cPAC learning is problematic in the agnostic case, when we cannot guarantee a computable Empirical Risk Minimizer (ERM). In computable universal online learning, proper is problematic already in the realizable case. In the realizable cPAC, every $RER$ class of finite VC dimension is properly cPAC learnable in the realizable case (see Agarwal et al 2020). This is because in the realizable setting, any $RER$ class has a computable ERM. This can be contrasted with the universal online learning setting, where even a computable ERM is not always enough to construct a computable learner. This observation (Proposition 1) is one of the main contributions of our paper.

---

> > ### Comment · Reviewer_K2dL · 2025-08-05
> >
> > Thank you for your explanations.
> >
> > I have changed my score to a 5, given that (i) the authors are committing to changing the presentation of the paper to make it more readable to a wider NeurIPS audience and to disambiguate definitions as mentioned above, and (ii) a closer read of the paper on my end and the authors' rebuttal, which I found clarifying.
> >
> > I would like to echo reviewer b4op's comments on the importance of computability. We have seen in the last 5 years have computability requirements been studied in various learning settings where learners are usually seen as functions (vs algorithms). I strongly believe this is something worth studying in its own right. I do agree however that this submission may be more relevant to COLT, ALT, STOC, etc., but I think it still has its place at NeurIPS.

---

### Note · Authors · 2025-08-12

Once again we want to thank each reviewer for the time devoted to reading and evaluating our paper. We are also grateful for the discussion during rebuttal, which helped to solve out the initial misunderstanding. Currently, three out of four reviewers support the acceptance of the paper, with the fourth evaluating it as borderline reject. To summarize, the reviewers noted that the paper:
1. addresses a substantial gap in the foundational topic that has been actively researched in the last 5 years.
2. is technically solid.
3. is generally clear and would improve further after the rebuttal discussion.

The fourth reviewer argued that computability is a notion that is too coarse to be practically relevant but noted the fundamental nature of the paper’s topic. Other reviewers added that the notion is still relevant as the minimal constraint of algorithmic effectivity and pointed to the fact that computable learning has been an active topic in machine learning theory for the last few years.

---

### Decision · Program_Chairs · 2025-09-17

**Decision:**

Accept (poster)

**Comment:**

The paper studies the problem of defining computable learners for online learning, under universal quantification (i.e., we require only a finite number of mistakes on every realizable data sequence, not necessarily bounded over all such sequences).  Both universal online learning and computable online learning have been studied actively in recent years.  One might be tempted to guess that computable universal online learning might be a simple mash-up of these two theories, with all the same equivalences and separations.  But surprisingly, this paper shows it's not so simple.  Unlike uniform online learning, this work shows there exist recursively enumerable classes that are universally online learnable but not computably universally online learnable.  They also give an extension to the agnostic case, where they show there exist realizable-case computably universally online learnable classes which are not agnostic-case computably universally online learnable (though the agnostic-to-realizable reduction continues to work in the case of recursively enumerable classes).  They further completely characterize the agnostic-case computably universally learnable classes.  They also characterize properly learnable classes.  The characterizations reflect an important connection to topological closures of concept classes.  These results reveal an interesting landscape of scenarios, beyond a mere extension of the theory of computable uniform online learning.

The reviewers agree that the results are novel, and align with recent topics of interest in learning theory.

Some reviewers (though not others) found issues in the presentation, leading to much confusion about the setting.  The authors have indicated they will implement appropriate changes to address these.

Another review notes the paper would be enhanced by the addition of natural examples illustrating the theory (e.g., perhaps a higher-dimensional example such as the discrete positive-halfspaces example in Bousquet et al. 2021).